# Unselfish traits and social decision-making patterns characterize six populations of real-world extraordinary altruists

Shawn A. Rhoads [1] ✉, Kruti M. Vekaria[1], Katherine O'Connell [1], Hannah S. Elizabeth[1], David G. Rand [2], Megan N. Kozak Williams[3] & Abigail A. Marsh [1]

Acts of extraordinary, costly altruism, in which significant risks or costs are assumed to benefit strangers, have long represented a motivational puzzle. But the features that consistently distinguish individuals who engage in such acts have not been identified. We assess six groups of real-world extraordinary altruists who had performed costly or risky and normatively rare (<0.00005% per capita) altruistic acts: heroic rescues, non-directed and directed kidney donations, liver donations, marrow or hematopoietic stem cell donations, and humanitarian aid work. Here, we show that the features that best distinguish altruists from controls are traits and decision-making patterns indicating unusually high valuation of others' outcomes: high Honesty-Humility, reduced Social Discounting, and reduced Personal Distress. Two independent samples of adults who were asked what traits would characterize altruists failed to predict this pattern. These findings suggest that theories regarding self-focused motivations for altruism (e.g., self-enhancing reciprocity, reputation enhancement) alone are insufficient explanations for acts of real-world self-sacrifice.

Almost seven decades ago, Maslow lamented psychologists' focus on negative aspects of human interpersonal relationships, writing that, "kindness, generosity, benevolence, and charity have too little place in the social psychology textbooks," and asking, "Where are the researches on unselfishness?"[1]. Although insightful research on this topic has emerged in recent years, the underpinnings of real-world acts of extraordinary altruism—for example, non-directed organ donations, heroic rescues, and risky humanitarian aid work—remain a puzzle[2–4]. Proximal or distal self-enhancing goals such as reciprocity, cooperation, norm conformity, and/or reputation-based motivations that drive many common forms of prosocial behavior do not adequately explain instances of altruism in which altruists assume significant concrete risks or costs to benefit anonymous strangers[2–4]. A variety of factors have been posited to drive such acts, including empathy[5–7], agreeableness[8], or temperamental boldness[9]. But whether real-world

extraordinary altruists are actually distinguished by these factors has not been empirically tested. We thus sought to assess adults who had previously engaged in any of six forms of extraordinary altruism. We predicted that extraordinary altruists would primarily be distinguished not by these factors (e.g., agreeableness, boldness, empathy), but by high levels of unselfishness[10–13]—that is, unusually high subjective valuation of the welfare of others relative to the self. In a series of five studies, we found that altruists are best distinguished from typical adults by their unusually unselfish traits and preferences. Although Maslow might have predicted this outcome, we also found that members of the general population did not.

Several lines of evidence point to extraordinary altruism reflecting stable unselfish traits. Stable trait-level tendencies that bias ecological and social behaviors in consistent ways are observed across organisms of many species[14–17] and are most predictive of behavioral outcomes in

[1]Georgetown University, Washington, DC, USA. [2]Massachusetts Institute of Technology, Cambridge, MA, USA. [3]Linfield University, McMinnville, OR, USA. ✉e-mail: sr1209@georgetown.edu

novel contexts lacking strong norms[18]. In such situations, no information about how to behave adaptively is known, such that situational constraints on behavior are weaker and individual differences are most likely to be revealed[18]. The rarity of extraordinary acts like heroic rescues and altruistic organ donations renders contexts in which these acts occur novel and lacking strong norms by default, and thus particularly likely to correspond to dispositional variation. Some evidence has linked one form of extraordinary real-world altruism (non-directed kidney donation) to reduced psychopathy and increased valuation of the welfare of socially distant others (decreased social discounting)[19], consistent with trait unselfishness. But other research suggests altruism primarily reflects traits other than unselfishness. Self-reported or laboratory-measured prosociality is often associated with trait agreeableness as measured by five-factor inventories[8,20], but these findings may be confounded by agreeableness promoting conformity to norms and expectations, which are highly salient in self-report and laboratory measures of prosociality[21,22]. Other evidence suggests real-world helping and even heroism are primarily driven by contextual factors or by traits such as boldness, norm-insensitivity, or even psychopathology[9,23-25].

Prior efforts to identify traits that reliably characterize altruists may have been hindered by small or constrained samples of altruists (e.g., only first responders or organ donors). In addition, commonly used global personality inventories may not adequately capture traits relevant to unselfishness[26]. For example, attributes linked to virtue and morality were deemed "insufficiently psychological" and intentionally excluded from the lexical terms used to develop early five-factor personality scales—possibly reflecting historical assumptions of universal and invariant selfishness[27].

We thus aimed to assess unselfish personality features in a sample of adults who had engaged in one of six forms of rare, real-world extraordinary altruism: heroic rescuers, non-directed and directed kidney donors, liver donors, marrow or hematopoietic stem cell donors, and humanitarian aid workers. We predicted that these groups of real-world altruists would be best distinguished from controls by unselfish traits and social decision-making patterns. In addition, we assessed perceptions of real-world altruists' dispositional traits among two independent samples that are demographically representative of American adults (208 participants in an exploratory study and 201 participants in a pre-registered confirmatory study); hypothesizing that altruism would be perceived as reflecting predominantly traits other than unselfishness.

We first assessed how altruists differ from typical people in terms of major personality dimensions, including those relevant to trait unselfishness and other traits previously linked to altruism (fearlessness, impulsive decision-making). Our battery of measures included both a comprehensive six-factor assessment of personality, as well as measures of more specific traits, including risk-taking, risk perception, cognitive reflection, empathy, and psychopathy. We administered this battery to 554 participants, who included six populations of rare real-world extraordinary altruists and controls. Extraordinary altruism was defined as acts that are normatively very rare (<0.00005% annual prevalence rate per capita in United States; see Supplementary Information, Table S1), and thus unlikely to reflect learned behaviors or norms, and that are aimed at benefiting the recipient at some significant risk or cost to the altruist. Although the extreme rarity of these acts presents recruitment challenges, operationalizing altruism in terms of stringently defined real-world acts minimizes social desirability and norm-adherence motives—which may confound laboratory-elicited altruism—and obviates ethical and practical considerations that prevent genuinely costly altruism from being elicited in the laboratory.

Analyses of 347 altruists included heroic rescuers ($N = 27$; annual U.S. prevalence: 0.00000024%), who had received a Carnegie Medal for "risking their lives to an extraordinary degree saving or attempting

to save the lives of others"[28]; non-directed (altruistic) kidney donors ($N = 132$; prevalence: 0.00000089%) who had donated a kidney to an anonymous stranger[29]; directed kidney donors ($N = 68$; prevalence: 0.00000906%) who had donated a kidney to a specified other person[29]; liver donors ($N = 12$, 8 directed and 3 non-directed; prevalence: 0.00000113%)[29]; bone marrow or hematopoietic stem cell donors ($N = 55$; prevalence: 0.00001530%) who had donated bone marrow to an unspecified stranger[30]; and humanitarian aid workers ($N = 53$; prevalence: 0.00000128%) who had performed work for organizations such as Médicins Sans Frontieres (Doctors Without Borders; participants included 41 North American respondents as well as 12 from other locales)[31]. Altruists were recruited in partnership with local and national organizations including the Carnegie Hero Fund, several living organ donation organizations (including Transplant Village and the Washington Regional Transplant Program), the National Marrow Donor Program®/Be the Match®, and Doctors Without Borders. Analyses also included 207 control participants recruited from the local community through flyers and postings on Research Match. Extensive screening confirmed they did not meet criteria for any of the above altruist categories (see Supplementary Information, Table S2 for demographics by group). We also conducted supplemental analyses of 5,000 bootstrap samples drawn from a large population of 347,192 adults (including 158,130 U.S. adults)[32,33], who were matched to altruists on age and sex.

In this work, we show that extraordinary altruists, including altruistic organ and marrow donors, heroic rescuers, and humanitarian aid workers, are best distinguished from typical adults by their unusually unselfish traits and decision-making patterns (including high honesty-humility, reduced social discounting, and reduced personal distress), indicating that altruists share unusually high valuation of others' welfare. Survey results from two independent samples of participants show these findings are not self-evident, as typical adults do not accurately predict what traits actually distinguish extraordinary altruists.

## Results

### Unselfish traits characterize altruists

To assess broad personality dimensions, including unselfishness, we administered the six-factor HEXACO assessment of personality to all participants. This inventory provides broad coverage of the personality space and less redundancy between dimensions[34]. In addition to the dimensions similar to those captured by five-factor inventories (emotionality, extraversion, agreeableness, conscientiousness, and openness to experience) this inventory includes honesty-humility[26,35], which specifically captures values and behaviors related to seeking personal gain at others' expense (exploitation)—in other words, relative valuation of outcomes for the self versus others[36]. Low levels of this trait uniquely predict behaviors that benefit the self at a cost to others, including greed, cheating, manipulativeness, and aggression[36]. We thus predicted that high Honesty-Humility would best discriminate real-world extraordinary altruists, whose behavior benefits others at a cost to the self, from controls. Participants also completed measures of other traits previously found to correspond to altruism, including cognitive reflection[37], risk-taking and risk-perception[38,39], empathy[40], and psychopathy[41]. The battery required approximately 60 min to complete, as part of an extensive screening that collected data on personality characteristics, demographic details, and measures assessing mental health and MRI safety to assess eligibility for future laboratory and neuroimaging studies. Participants were compensated $20 for completing the battery of measures.

We conducted a series of multiple linear regression analyses to assess how altruists were distinguished from controls across these measures (including subscale scores). Controls were recruited to approximately match altruist demographics. This recruitment strategy achieved approximate demographic matching in sex (60.81% female

altruists in comparison to 64.73% female controls; $\chi^2(1) = 0.85$, $p = 0.356$) and household income (see Supplementary Information, Table S2, for percentage breakdown; $\chi^2(8) = 14.93$, $p = 0.061$). However, because groups were recruited concurrently, altruists were older (mean = 44.06 years, SD = 12.50) in contrast to controls (mean = 37.71 years, SD = 9.07), $T(552) = 6.37$, $p < 0.001$) and less educated (73.49% altruists received a college education in contrast to 87.92% controls, $\chi^2(1) = 16.22$, $p < 0.001$) (see Supplementary Information, Table S2). Therefore, all regressions controlled for age and sex (among all participants), as well as income and education (among participants with available data), which varied across altruistic groups.

Results confirmed that the variable that most consistently distinguishes real-world extraordinary altruists from typical adults is honesty-humility, with 5 of the 6 groups of altruists scoring higher on this trait than controls (Fig. 1; see Supplementary Information, Tables S3a–f); the exception were liver donors ($N = 12$), whose mean scores were similar to other altruistic groups but for whom small sample size (owing to the extreme rarity of living liver donations) limited statistical power. Four groups (directed and non-directed kidney donors, heroic rescuers, and humanitarian aid workers) also scored lower on the Personal Distress subscale of the Interpersonal Reactivity Index, which indexes tendencies to experience self-focused distress in emergencies. No more than 3 of the 6 groups of altruists differed from controls in terms of any other variable (Fig. 2; see Supplementary Information, Figures S1a–d, Tables S3–8). We conducted follow-up analyses that also controlled for age, sex, education, and household income (for all participants with available data; $N = 533$) and that yielded similar results (see Supplementary Information, Tables S3–8).

Because our community sample of controls was not perfectly matched to altruists due to a variety of factors (for example, altruists are often selected based on factors related to age, sex, and health, and each altruistic group varied widely in their demographic makeup relative to others; see Discussion for further information regarding these limitations), we also sought to confirm whether our finding that Honesty-Humility is the dimension of the HEXACO that most reliably distinguishes real-world extraordinary altruists would replicate in an independent control dataset. To accomplish this, we acquired data from a large population of 347,192 participants who completed the same HEXACO items measured in the present study[32,33]. We stratified the international dataset by country (United States), age (quantile split), and sex, and randomly drew 5,000 bootstrap samples without replacement that were matched to the full altruistic sample on country, age, and sex. We then compared our altruist sample ($N = 347$) and our initial control sample ($N = 207$) to this new distribution of 5,000 mean scores for each of the HEXACO personality dimensions. We found that our control sample did not differ from the distribution of bootstrap means for any of the HEXACO dimensions. Even more importantly, we replicated our finding that Honesty-Humility was the only dimension of the HEXACO for which altruists' mean scores (and 95% confidence intervals) did not overlap with the distribution of bootstrap means ($p < 0.001$) (see Supplementary Information, Fig. S2).

### Altruists assign greater value to distant others' welfare
Following evidence that unselfish self-reported traits best discriminate altruists from controls, we inquired whether altruists reliably demonstrate unusually unselfish decision-making patterns in a controlled social discounting task indexing selfish and unselfish preferences. Social discounting indexes the subjective value of resources as a function of whether they are kept (selfish choices) or shared with others (generous choices), and how the subjective value of shared resources declines for progressively more socially distant recipients[42,43]. It is a robust phenomenon, with respondents across settings and cultures reliably showing more selfish choices as the social distance between respondents and beneficiaries increases, following a

hyperbolic function[19,44–47]:

$$v = \frac{V_0}{1 + k*N} \qquad (1)$$

where $V_O$ represents the intercept or undiscounted value of the reward, $k$ represents degree of discounting (discounting rate), $N$ represents social distance (for example, 1 representing the closest social relationship such as a spouse or child and 100 representing a stranger), and $v$ represents the amount willing to forgo for each social other. As $N$ increases, the resources individuals are willing to forgo ($v$) typically decrease in a hyperbolic fashion governed by the parameter, $k$, which represents the rate by which the function decays across social distance.

In a subset of our original sample ($N = 275$, including 217 altruists and 58 controls; see Supplementary Information, Table S9), we assessed social discounting using a validated online task. At the outset, participants saw the following instructions: "Imagine that you have made a list of the 100 people closest to you in the world… The person at number one would be someone you know well and is your closest friend or relative. The person at #100 might be someone you recognize and encounter but perhaps you may not even know their name." Then, seven randomly presented blocks inquiring about sharing resources with person number 1, 2, 5, 10, 20, 50, or 100 on the list followed. Within a block, participants made 9 binary choices to keep or forgo a certain amount of money for the specified person. Indifference points were estimated via logistic function to determine participants' "amount willing to sacrifice" ($v$) for each social other ($N$).

Results of a hyperbolic mixed-effects model (see Methods) controlling for age and sex indicated that social discounting ($logk$) discriminated four types of altruists from controls, with all groups discounting less on average than controls (Table 1, Fig. 3). Specifically, our model revealed that liver donors ($p = .015$, $coefficient = -1.93$, $CI_{95\%} = [-3.48, -0.38]$), non-directed kidney donors ($p = 0.004$, $coefficient = -1.33$, $CI_{95\%} = [-2.24, -0.43]$), humanitarian aid workers ($p = 0.015$, $coefficient = -1.17$, $CI_{95\%} = [-2.10, -0.23]$), and directed kidney donors ($p = 0.029$, $coefficient = -.97$, $CI_{95\%} = [-1.84, -0.10]$) discount less than typical adults. Post-hoc linear regressions examined willingness to forgo resources relative to controls at seven social distances and results are reported in Supplementary Information, Tables S10a–g. Follow-up analyses controlled for age, sex, education, and household income (for all participants with available data; $N = 264$) and yielded similar results (see Supplementary Information, Tables S10a–g).

To investigate the relationship between discounting and Honesty-Humility, we entered a group×Honesty-Humility interaction term at level 2 in our hyperbolic mixed-effects model. Results revealed that liver donors, non-directed kidney donors, and directed kidney donors discount significantly less than controls as Honesty-Humility trait scores increases. Generally, discounting rates decreased as Honesty-Humility increased (Fig. 4, Table 1), consistent with increased Honesty-Humility supporting reduced social discounting rates among altruists. Follow-up analyses included age, sex, education, and household income as covariates (for all participants with available data; $N = 264$) and yielded similar results (see Supplementary Information, Table S11).

### Unselfish Traits and Social Discounting Predict Altruism
To account for possible shared variance among all characteristics measured, we next performed a classification analysis aimed at identifying which variables best contributed to predicting the probability that a respondent is an altruist (grouping altruistic groups together) versus control when all variables were considered simultaneously. Data from 275 observations were randomly split 80/20 into training/testing partitions. Using a penalized logistic classifier with L1 regularization (Least Absolute Shrinkage and Selection Operator; LASSO),

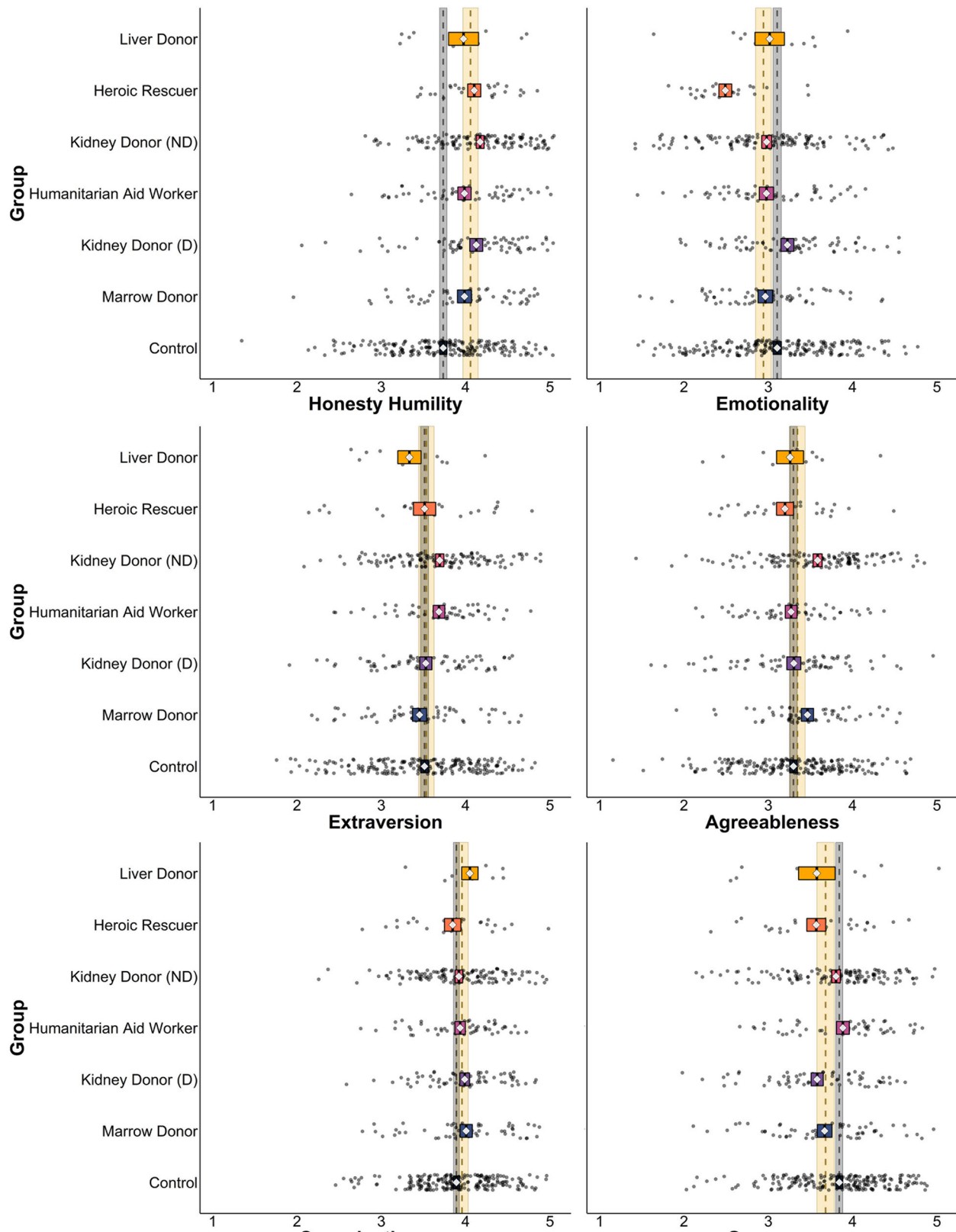

**Fig. 1 | HEXACO personality traits across six groups of altruists and controls.**
Gray dashed line and ribbon represent control group mean and 95% CIs ($N = 206$), yellow dashed line and ribbon represent combined altruist group mean and 95% CIs ($N = 347$), diamonds represent means of individual groups, box widths represent

95% CIs. Multivariate regressions compared all altruistic groups against controls simultaneously using two-sided tests (in lieu of separate tests for which corrections for multiple comparisons would be appropriate).

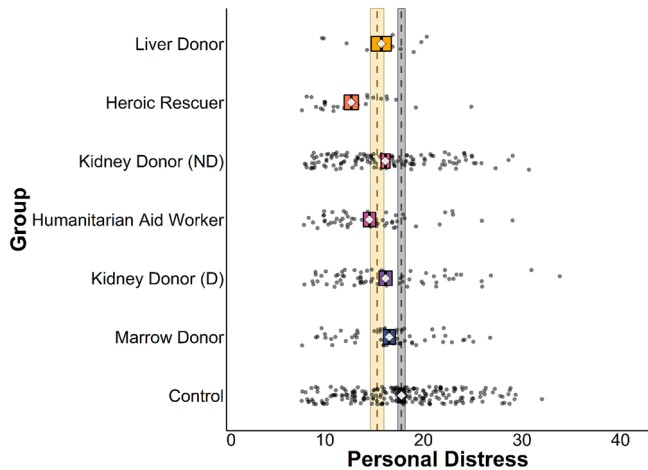

**Fig. 2 | Personal Distress across six groups of altruists and controls.** Gray dashed line and ribbon represent control group mean and 95% CIs (*N* = 207), yellow dashed line and ribbon represent combined altruist group mean and 95% CIs (N = 347), diamonds represent means of individual groups, box widths represent 95% CIs. Multivariate regressions compared all altruistic groups against controls simultaneously using two-sided tests (in lieu of separate tests for which corrections for multiple comparisons would be appropriate).

which shrinks coefficients of less contributive variables toward zero, we trained a model using 5-fold cross-validation to optimally classify altruists versus control using all 25 variables reported above (see Methods). The model selected: social discounting rate (*logk*), Honesty-Humility (HEXACO), Openness to Experience (HEXACO), Social Risk-taking (DOSPERT), and Personal Distress (IRI). Using these variables, we were able to achieve above-chance classification accuracy in our testing partition (AUC = .76; see Supplementary Information, Figure S3). Using all available data from 264 participants (see Supplementary Information Table S12), a binary logistic model using these selected variables, and controlling for age, sex, education, and household income, revealed that social discounting rate, honesty-humility, and personal distress explained the most variance in predicting altruistic group (Table 2). For every unit increase in honesty-humility score, participants were 1.84 times more likely to be altruists; for every unit increase in discounting rate, participants were 0.20 times less likely to be altruists; and for every unit increase in Personal Distress score, participants were 0.09 times less likely to be altruists. Notably absent were predictors linked to altruism in prior research, including self-reported empathy[5–7], agreeableness[8] (although note that the agreeableness dimension of the HEXACO is not identical to this dimension in five-factor inventories), or temperamental risk-perceptions/boldness[9]. Note also that because the base rate of these altruistic behaviors is so low even an 84% increase in that base rate still yields a low value.

All means and standard deviations of trait and behavioral measures among altruists and controls are reported in Supplementary Information (Table S13). Bivariate correlations among variables are also reported in Supplementary Information (Table S14).

## Lay beliefs about altruists and other rare groups
We also aimed to assess whether beliefs about of altruists in the general population were consistent with the personality traits and decision-making patterns we observed among actual altruists. We conducted this assessment in an exploratory study (*N* = 208) and a pre-registered confirmatory study (*N* = 201; pre-registration link: https://osf.io/7t4qf/). Each study recruited a sample of American adults using a Qualtrics panel designed to match census-based population demographics in terms of sex, age, race and ethnicity, and education (see Supplementary Information, Table S15).

Procedures were similar across both studies. Participants were asked to consider how an individual representing each of the six groups of altruists we assessed compared to the average person in terms of altruism (i.e., how altruistic they perceived each to be), risk (i.e., how risky they perceived each type of altruism to be), and the six HEXACO dimensions (detailed explanations of all traits were provided for each study; see Supplementary Information, Table S16a, b).

All participants also completed a third-person social discounting task, which assessed their judgments about how an individual who had engaged in each of the six forms of altruism (as well an average person) would be likely to allocate resources for others (see Methods). These ratings directly represented beliefs about the "amount each target would be willing to forgo" (*v*$_{beliefs}$), such that estimated social discounting rates (*logk*$_{beliefs}$) could be calculated using the hyperbolic discounting model separately for each group for each participant.

Across both the exploratory and confirmatory studies, linear mixed-effects modeling revealed that respondents perceive all forms of extraordinary altruism to be both relatively altruistic and risky (see Supplementary Information, Table S17a-i). Both samples also judged altruists as differing from the average person in terms of five out of six HEXACO personality dimensions. Estimated social discounting rates corresponding to judgments about how much an individual is willing to forgo for various social others (*logk*$_{beliefs}$) were shallower for all altruistic groups relative to those estimated about the average person. Thus, respondents correctly viewed altruists as more willing to forgo resources for others at varying social distances (social discounting)—linking altruism to unselfish behavior. However, they did not specifically associate extraordinary altruism with traits related to unselfishness. Rather, they associated extraordinary altruism with undifferentiated traits broadly construed as positive, thus predicting 1 out of 6 personality dimensions correctly (see Supplementary Information, Figure S4a-e).

To explore whether this prediction error was specific to altruism, or whether it was simply a function of extraordinary altruists' rarity, we also recruited two independent samples drawn from equally rare populations (*N* = 28) that are not defined by altruism. They included a sample of extreme athletes (BASE jumpers)[48–51] and former contestants in the national Miss America pageant. All completed the battery of HEXACO-PI-R items. These groups were selected because their prevalence rate can be estimated and are similar to that of our altruistic groups. There have been roughly 2500 BASE Jumpers worldwide since 1981 as estimated by basenumbers.org as of August 2022 (prevalence assuming all are alive today: 0.0000075%)[52], there are 51 Miss America contestants per year (prevalence assuming all are alive today since the first pageant in 1921: 0.000015%). Furthermore, the annual fatality rate of BASE jumping is higher than that of organ donation, estimated to be 1 death for every 60 jumpers and the serious injury rate (requiring hospital care) as 0.2–0.4% per jump (i.e., a 5- to 16-fold risk for death or injury when compared with skydiving)[51].

Using the same bootstrapping procedure described above, we then compared each of these new rare groups to a distribution of 5,000 bootstrap samples that were matched to each group on country (United States), age (quantile split), and sex (female only for Miss America contestants). BASE jumpers on average scored lower in emotionality, higher in agreeableness, and lower in openness to experience compared to the matched distribution of bootstrap means (see Supplementary Information, Figure S5a). We found that Miss America contestants on average scored lower in honesty-humility, higher in extraversion, and higher in conscientiousness compared to the matched distribution of bootstrap means (see Supplementary Information, Figure S5b).

We then tested whether the general population from the second sample of participants (N = 201) would uniquely link each group to these characteristics. We found that the general population was broadly accurate at predicting Miss America contestants' traits

**Table 1 | Hyperbolic mixed-effects model results for group differences in social discounting**

| N = 275 | Estimate | SE | 95% CI | T | p | Estimate | SE | 95% CI | T | p |
|---|---|---|---|---|---|---|---|---|---|---|
| **Level 1** | | | | | | | | | | |
| v0 | 85.501[c] | 0.482 | [84.557, 86.446] | 177.513 | <0.0001 | 78.481[c] | 0.871 | [76.772, 80.191] | 90.064 | <0.0001 |
| Social discounting rate (logk) | −2.796[c] | 0.620 | [−4.013, −1.580] | −4.509 | <0.0001 | 2.168[b] | 0.739 | [0.719, 3.617] | 2.935 | 0.003 |
| **Level 2** | | | | | | | | | | |
| Heroic rescuer | −0.722 | 0.745 | [−2.183, 0.738] | −0.970 | 0.332 | −4.009 | 2.585 | [−9.080, 1.062] | −1.551 | 0.121 |
| Humanitarian aid worker | −1.165[a] | 0.478 | [−2.104, −0.227] | −2.435 | 0.015 | −0.145 | 1.267 | [−2.631, 2.341] | −0.114 | 0.909 |
| Kidney donor (D) | −0.971[a] | 0.443 | [−1.840, −0.102] | −2.192 | 0.029 | −4.959[c] | 1.022 | [−6.964, −2.954] | −4.852 | <0.0001 |
| Kidney donor (ND) | −1.333[b] | 0.461 | [−2.238, −0.429] | −2.890 | 0.004 | −5.428[c] | 1.203 | [−7.787, −3.069] | −4.514 | <0.0001 |
| Liver donor | −1.928[a] | 0.789 | [−3.477, −0.380] | −2.442 | 0.015 | −8.641[b] | 2.684 | [−13.905, −3.377] | −3.220 | 0.001 |
| Marrow donor | −0.633 | 0.499 | [−1.612, 0.345] | −1.269 | 0.205 | −0.225 | 1.187 | [−2.553, 2.103] | −0.190 | 0.85 |
| Sex (Female) | 0.285 | 0.320 | [−0.343, 0.913] | 0.891 | 0.373 | 0.51[c] | 0.131 | [0.253, 0.767] | 3.896 | <0.0001 |
| Age | −0.018 | 0.013 | [−0.042, 0.007] | −1.399 | 0.162 | −0.012[a] | 0.005 | [−0.021, −0.002] | −2.319 | 0.02 |
| Honesty humility (HH) | | | | | | −1.524[c] | 0.202 | [−1.921, −1.128] | −7.548 | <0.0001 |
| Heroic rescuer × HH | | | | | | 0.979 | 0.643 | [−0.282, 2.240] | 1.523 | 0.128 |
| Aid worker × HH | | | | | | −0.147 | 0.342 | [−0.819, 0.524] | −0.430 | 0.667 |
| Kidney donor (D) × HH | | | | | | 1.111[c] | 0.268 | [0.586, 1.637] | 4.146 | <0.0001 |
| Kidney donor (ND) × HH | | | | | | 1.181[c] | 0.308 | [0.576, 1.785] | 3.831 | <0.0001 |
| Liver donor × HH | | | | | | 1.802[b] | 0.661 | [0.506, 3.098] | 2.727 | 0.006 |
| Marrow donor × HH | | | | | | 0.009 | 0.315 | [−0.608, 0.625] | 0.027 | 0.978 |

Note. Groups are coded as indicator variables relative to the control group. Fixed-effects coefficients are unstandardized. SE indicates the standard error. 95% CI indicates lower/upper limits of the confidence interval. Because social distance (N) was centered at $N = 1$, $v_0$ represents the amount willing to forgo for $N = 1$.
[a] indicates $p < 0.05$,
[b] indicates $p < 0.01$,
[c] indicates $p < 0.001$. *P*-values are two-tailed.

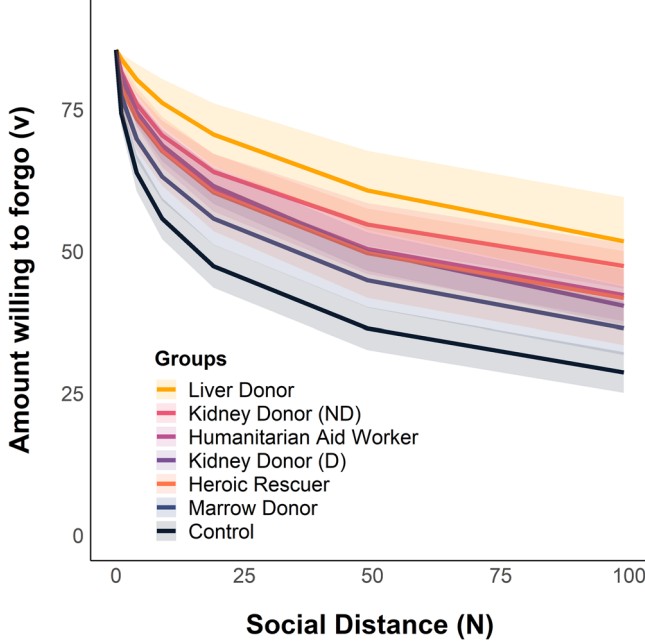

**Fig. 3 | Hyperbolic social discounting across six groups of altruists and controls.** Labels are ordered in descending order of mean discounting slope across groups (logk). Lines represent the mean amount willing to forgo for each group as predicted by the hyperbolic model. Ribbons represent the 95% CIs. Multivariate mixed-effects regressions compared all altruistic groups against controls simultaneously using two-sided tests (in lieu of separate tests for which corrections for multiple comparisons would be appropriate).

(correctly predicting 5 out of 6 personality dimensions), but were less accurate at predicting BASE Jumpers' traits (correctly predicting 2 out of 6 personality dimensions) (see Supplementary Information, Table S18a-h). This finding provides preliminary evidence that the

general population's prediction errors for altruists (who they misjudged on 5 out of 6 traits) might be specifically related to misconceptions of altruism and not simply reflect altruistic groups being so rare. This finding also suggests that the general population's responses regarding altruists did not result from a broad decision-making error because participants' beliefs were broadly accurate for Miss America contestants.

Finally, we tested how closely perceptions of extraordinary altruists across two representative samples of the general population correspond to actual features of extraordinary altruists using representational similarity analysis (RSA)[53]. This analysis allowed us to assess how the relative degree of similarity in perceived traits among real-world altruists corresponds to the relative degree of similarity among altruists' actual traits. We conducted pairwise comparisons of perceived and actual models of altruism that considered the six dimensions comprising the HEXACO. We broadly found high similarity within the representational structure characterizing altruists' actual traits and within the structure characterizing perceptions of altruism (see Supplementary Information, Figure S6). While perceived representations of altruists were consistent across the exploratory and confirmatory sample, typical adults' perceived representations of altruists failed to capture consistent or meaningful variance in representations of actual altruism (see Supplementary Information, Figure S7a-c).

## Discussion

We assessed the personality structure of six groups of rare, real-world extraordinary altruists in an effort to identify traits that generally characterize them. The traits that best distinguished altruists from typical adults are those most closely linked to unselfishness: increased Honesty-Humility, reduced Social Discounting, and reduced Personal Distress (self-focused distress in emergencies). In a series of linear regression analyses and an ensuing logistic classification analysis, we found these traits best discriminated altruists from controls. We then replicated this finding using a larger sample of controls who were

matched to altruists on age and sex. By contrast, altruists were not distinguished by other traits previously linked to altruism, including self-reported empathy, agreeableness (as measured by the HEXACO), boldness, and risk insensitivity. But, despite empirical evidence specifically linking real-world altruism and unselfishness, two independent representative samples of American adults did not specifically associate extraordinary altruism with traits related to unselfishness, but rather with undifferentiated traits broadly construed as positive but not closely related to actual altruism, including high extraversion, high agreeableness, and high conscientiousness. (Furthermore, this prediction error seemed to be relatively specific to altruism, as the general population was more accurate at predicting traits characterizing similarly rare groups who were not defined by altruistic actions.) Using RSA, we observed poor correspondence between representational structure characterizing altruists' actual traits and the structure characterizing perceptions of altruism across both representative samples. Together, these findings support a broadly reliable relationship between stringently defined real-world altruistic behavior and

unselfish traits and decision-making patterns—an association the average person may not expect.

That extraordinary altruists are consistently distinguished by a common set of traits linked to unselfishness is particularly noteworthy given the differences in the demographics of the various altruistic groups we sampled and the differences in the forms of altruism they have engaged in—from acts of physical heroism to the decision to donate bone marrow. This finding replicates and extends findings from a previous study[19] demonstrating that extraordinary altruists show heighted subjective valuation of socially distant others. In addition, our results are consistent with a recent meta-analysis of 770 studies[54] finding a strong and consistent relationship between Honesty-Humility and various forms of self-reported and laboratory-measured prosociality. Coupled with findings that low levels of unselfish traits (e.g., low Honesty-Humility, high social discounting) correspond to exploitative and antisocial behaviors such as cheating and aggression[36,55], these results also lend support to the notion of a bipolar caring continuum along which individuals vary in the degree to which they subjectively value (care about) the welfare of others[56–58]. They further suggest altruism—arguably the willingness to be voluntarily "exploited" by others—to be the opposite of phenotypes like psychopathy that are characterized by exploiting others[59]. These traits may best predict behavior in novel contexts lacking strong norms[18], particularly when decisions are made rapidly and intuitively[14,57]. Notably, people who are higher in prosociality are more likely to participate in psychological research to begin with[60]—thus the observed differences between altruists and controls may be underestimates (i.e., population-level differences may be larger).

These findings indicate social discounting patterns are meaningful, replicable, non-self-report-based indices of individual variation in ecologically valid, extraordinary forms of altruism. In that social discounting decisions directly reflect variation in the subjective value of outcomes for various others versus the self, they are valuable for understanding the basis of extraordinary altruism—making large sacrifices or taking significant risks for more distant others is inherently more extraordinary because it is much rarer across all populations. In that subjective value is typically defined as the internal value a stimulus has to motivate choices and behavior[61], these findings may help to understand motivations underlying extraordinary altruism. Although the neural mechanisms that determine the subjective value of outcomes for the self are well delineated[62–65], relatively less work has focused on processes underpinning the subjective valuation of outcomes for others[45,66–68]. Previous work suggests a critical role for

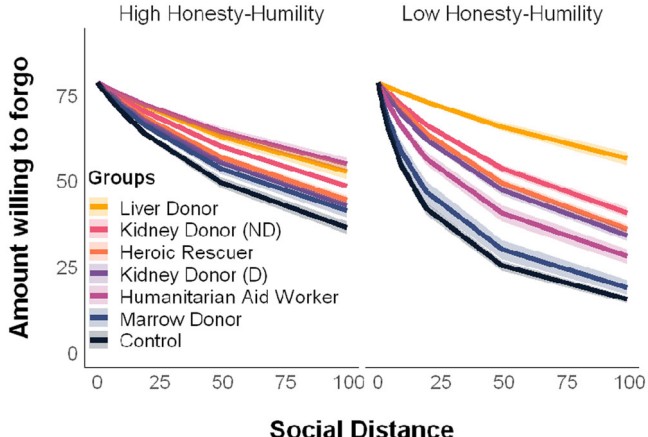

**Fig. 4 | Hyperbolic social discounting across six groups of altruists and controls by honesty-humility scores.** Honesty-Humility scores are mean split for visualization purposes. Labels are ordered in descending order of mean discounting slope across groups (logk). Lines represent the mean amount willing to forgo for each group as predicted by the hyperbolic model. Ribbons represent the 95% CIs. Multivariate mixed-effects regressions compared all altruistic groups against controls simultaneously using two-sided tests (in lieu of separate tests for which corrections for multiple comparisons would be appropriate).

## Table 2 | Binary logistic model results classifying altruists versus controls

| N = 264 | Odds ratio (OR) | 95% CI (OR) | Log Odds Ratio (LOR) | SE (LOR) | 95% CI (LOR) | Z | p |
|---|---|---|---|---|---|---|---|
| Intercept | 42.235[a] | [1.273, 1656.4] | 3.743 | 1.819 | [0.160, 7.326] | 2.057 | 0.04 |
| Social discounting rate (logk) | 0.801[b] | [0.677, 0.944] | −0.222 | 0.084 | [−0.388, −0.056] | −2.631 | 0.009 |
| Honesty-humility | 1.837[a] | [1.021, 3.363] | 0.608 | 0.302 | [0.013, 1.204] | 2.013 | 0.044 |
| Openness to experience | 0.598 | [0.333, 1.046] | −0.514 | 0.290 | [−1.086, 0.058] | −1.769 | 0.077 |
| Risk-taking (Social) | 0.705 | [0.460, 1.053] | −0.349 | 0.211 | [−0.764, 0.066] | −1.657 | 0.098 |
| Personal distress | 0.911[b] | [0.848, 0.977] | −0.093 | 0.036 | [−0.164, −0.023] | −2.602 | 0.009 |
| Age | 1.036[a] | [1.004, 1.071] | 0.035 | 0.016 | [0.003, 0.067] | 2.167 | 0.03 |
| Sex (Female) | 0.830 | [0.390, 1.727] | −0.186 | 0.378 | [−0.929, 0.558] | −0.492 | 0.623 |
| College education | 0.819 | [0.313, 1.956] | −0.199 | 0.462 | [−1.109, 0.711] | −0.431 | 0.666 |
| Income (≥$90k) | 1.027 | [0.509, 2.060] | 0.027 | 0.355 | [−0.673, 0.726] | 0.075 | 0.94 |

*Note* 95% CI indicates lower/upper limits of the confidence interval. Intercept corresponds to the mean for male controls at average age who never completed a four-year degree and household earns < $90,000.
[a]indicates *p* < 0.05,
[b]indicates *p* < 0.01,
*** indicates *p* < 0.001. *P*-values are two-tailed.

variation in the structure and function of brain regions such as the amygdala in extraordinary altruism[10]. Other studies in both humans and non-human primates implicate populations of neurons in the amygdala and regions within the medial prefrontal cortex (e.g., subgenual and rostral anterior cingulate cortex; ACC) as particularly relevant to valuation of others' outcomes. Discrete neurons have been identified in humans and non-human primates in a comparable region of rostral ACC, as well as amygdala, that specifically encode the value of outcomes for others[69–71]. In fMRI studies in humans, activity in subgenual ACC corresponds to prosocial learning rates, and patterns of activity in this region correspond to trait-level empathy[72]. Together, these findings suggest a conserved system by which social mammals can represent and calculate the subjective value of outcomes for others as distinct from the self, individual variation in which may support trait-level differences in unselfishness. It will be important to consider whether neural responses encoding the subjective valuation of others' outcomes support processes that distinguish altruists from typical individuals.

Despite the empirical link between altruism and unselfish traits and preferences, this association was not self-evident to two independent samples of representative American respondents, who did not link altruism specifically to characteristics related to unselfishness. Instead, participants in both our exploratory and confirmatory studies rated altruists relative to the average person as non-specifically more positive in terms of five out of six dimensions (i.e., higher honesty-humility, extraversion, agreeableness, conscientiousness, and openness to experience). This notably contrasts with observations of altruists' actual traits and behaviors—as measured using the HEXACO, they are not all-around "better" people who are generally more outgoing, nicer, more responsible, and more open-minded. They are only less selfish.

It is possible this discrepancy reflects a long history of academic and popular conceptualizations of altruism as resulting from various other personality traits—including fearlessness, agreeableness (as measured by five-factor inventories), conscientiousness, and mood enhancement goals[73]. Historically, traits linked to ethics and morality have been largely excluded from global measures of personality, with morally-linked traits sometimes assumed to be artifacts of selfish motives like social desirability goals[27]. This is despite the average person believing ethical and moral traits to be core to self and identity[74,75]. There is a pervasive tendency (including in our sample) to ascribe extreme moral actions to variations in conscientiousness or willpower rather than variation in core selfish preferences[76]—whereas our findings indicate the opposite may be true[77].

A few additional limitations of these findings should be considered. The recruitment of rare populations of real-world extraordinary altruists limited our sample sizes, particularly liver donors, who made up only 0.00000113% of the U.S. adult population at the time of recruitment (367 U.S. liver donations in 2017; 12 of which were non-directed, compared to 5,812 U.S. kidney donation in 2017; 258 of which were non-directed)[29]. Thus, the sample sizes were low, which might have affected the detection of true effects. It could be argued that the rarity of these forms of altruism may have made it more difficult for our population sample to predict what traits would distinguish altruists, although their greater accuracy in predicting the personality traits of BASE jumpers and Miss America contestants (who are similarly rare) contradicts this possibility. It should also be noted that altruists (particularly organ and marrow donors) are often selected based on factors related to age, sex, and health. In part because of this, altruist groups vary widely in their demographic makeup (see Supplementary Information, Table S1). This demographic variation could be considered an additional reason that the consistency observed in altruists' personality traits across the six groups is noteworthy. It is also a reason we did not recruit controls to match each individual group demographically, but rather to match the combined altruist sample. Our recruitment strategies explicitly sought controls of similar approximate demographics as altruists, and achieved approximate demographic matching in sex and income (see Supplementary Information, Table S2) although exact matching of age and education was not achieved during concurrent recruitment. Although recent work has found age and sex effects for Honesty-Humility[78], what group differences emerged (despite our efforts) were statistically controlled for in all analyses. Furthermore, our results are substantiated by supplemental analyses comparing altruists to bootstrapped control samples pooled from a large population sample ($N = 347,192$) that was matched to altruists on country, age, and sex. We found that our initial control sample not differ from the distribution of bootstrap means for any of the HEXACO factors, and also replicated the finding that Honesty-Humility was the only factor of the HEXACO in which altruists' mean (and 95% confidence intervals) did not overlap with the distribution of bootstrap means.

Temporal conclusions are also limited by the fact that altruistic behaviors were necessarily performed prior to testing. To draw stronger conclusions that unselfish traits precede altruistic acts in a similarly sized sample of altruists, we would need to collect longitudinal data on our measures in hundreds of thousands or millions of participants and follow them for 10 years or more until a small fraction of them performed real-world extraordinary acts of altruism (e.g., roughly 1 in every 5 million people is recognized as a heroic rescuer, and roughly 17 in every 1 million people is a directed kidney donor; see Supplementary Information Table S1). In lieu of this, we instead aimed to identify measures that distinguish individuals recruited because they had previously engaged in stringently defined real-world altruism. An alternate hypothesis could thus be that altruists' patterns of responding are the result of their altruistic acts. For example, they could have concluded based on their past behavior that they are unusually altruistic or compassionate. However, two lines of evidence argue against this possibility. First, altruists do not score highly on self-report altruism scales that are more straightforwardly related to altruism[22,79] (including empathic concern in the present study) and in interviews they consistently report that they are not especially altruistic or unusual in any way, but simply acted as anyone with the same information and opportunity would have[19,80]. This interview response is most consistent with the high Honesty-Humility scores observed in the present study. Second, data from our general population samples can be used as a proxy for the traits that the average person associates with extraordinary altruists. Our exploratory and confirmatory studies validate that the average person's beliefs about what traits distinguish people who have engaged in extraordinary acts of altruism do not closely match the traits that altruists actually report, which suggests altruists' measured traits and preferences do not likely simply reflect stereotypes about altruists that they possess.

The measures collected in the present study do not represent an exhaustive list of measures potentially related to real-world altruism. Although we did not pre-register these measures, they were selected to be as comprehensive as possible (the six-factor inventory) and to capture variables consistently linked to altruism in prior research, including Agreeableness[8] (again noting that the HEXACO agreeableness scale is not identical to five-factor inventory versions of this scale), Honesty-Humility[20,54], risk sensitivity[81], empathy[11,12], social discounting[19], and psychopathy[19]. We analyzed the data using multiple strategies for the sake of openness, replicability, transparency, and robustness of findings (and all data and code are available on the Open Science Framework for reproducibility). In prior work, we have found that extraordinary altruists are not distinguished from controls on measures of self-reported altruism[79], social value orientation, or altruistic punishment[22]. Note that our discounting task did not employ actual payouts but hypothetical ones; this choice was made following evidence for the validity of this design[19] and evidence that hypothetical and real payouts yield similar social discounting patterns[47,82]. Other

batteries of assessment should be administered among these altruistic populations to gain further insight into the characteristics related to extraordinary altruism, such as paradigms that map onto different types of prosocial decision-making according to divergent task and neural features[83]. These might include other economic games, those that examine moral decision-making[66,84], or those that investigate moral inferences about others[85].

Despite these limitations, the present results establish that unselfish traits and values reliably distinguish individuals who engage in real-world acts of altruism. In other words, extraordinarily unselfish acts can be interpreted as veridical indicators of globally unselfish traits and values. These findings do not contradict established theories regarding various other self-focused motivations for altruism such as expectations of reciprocity and reputation enhancement; rather, they indicate that these factors alone may be insufficient explanations for acts of real-world self-sacrifice.

## Methods

### Study 1a

All study 1 procedures were carried out in accordance with a protocol approved by the Institutional Review Board at Georgetown University in Washington, D.C., and all participants provided electronic written informed consent upon enrollment.

**Participants.** Altruistic and directed kidney donors, liver donors, and bone marrow or hematopoietic stem cell donors were recruited and invited to participate through communications via local and national transplant organizations (Transplant Village, National Marrow Donor Program; invitations sent to individuals who completed a donation) and our existing database of altruistic kidney donors. Humanitarian aid workers were recruited through online postings in Facebook groups of individuals affiliated with Doctors Without Borders, Médicins Sans Frontiere, as well as email communications sent out to staff and volunteers of International Medical Corps. Heroic rescuers were recruited through email invitations from the Carnegie Hero Fund to members of their internal database. Control participants were recruited from the local community, through flyers and postings on Research Match, and were included in the sample as long as they did not meet criteria for any of the above altruist categories.

Recruitment yielded a sample of 554 adult participants (345 female), ages 19–78. The sample included 347 altruists and 207 controls. The altruists were comprised of the following: bone marrow donors ($N = 55$), directed kidney donors ($N = 68$), humanitarian aid workers ($N = 53$), non-directed kidney donors ($N = 132$), heroic rescuers ($N = 27$), liver donors ($N = 12$; of these, directed, $N = 9$; non-directed, $N = 3$). One control participant did not complete the HEXACO and was list-wise excluded for analyses including this measure. 48 participants did not complete the cognitive reflection test (CRT) and were list-wise excluded for analyses including this measure (45 non-directed kidney donors, 3 heroic rescuers). The participants excluded from HEXACO and CRT analyses were included in all other analyses if they had complete data for the measures used in those analyses. In other words, all usable data were included in all analyses.

**Procedure.** Upon recruitment, participants completed an initial online screening, which included a demographics questionnaire, and the HEXACO short form consisting of 60 items[35]. They also completed a self-report measure of empathy (IRI, Interpersonal Reactivity Index)[40], a self-report measure of risk taking tendencies and perceptions (Domain-Specific Risk-Taking, DOSPERT)[38,39], a self-report measure on psychopathy (Psychopathic Personality Inventory-Short Form, PPI-SF)[41], and the CRT that measures fast and intuitive *versus* slow and reflective deliberation during decision-making[37]. Participants were compensated $20 for completing the battery of measures as part of an extensive screening that required approximately 60 min to complete,

and included all surveys, demographic questions, and measures assessing mental health and MRI safety to assess eligibility for future laboratory and neuroimaging studies.

**Linear regression.** To assess how individuals in each of the altruistic groups differed from typical adults across these 24 measures, we conducted a series of linear ordinary least squares regressions using the lme function in R version 3.6.3. In each model, we entered groups as indicator variables with the typical adults as the baseline group and included age and sex. For the CRT data, we split participants into two groups based on high accuracy (more deliberation, 2 or more correct responses) versus low accuracy (less deliberation, 1 or fewer correct responses) before running the linear regression.

We also conducted analyses that included education and household income as covariates. Education (completed four-year degree or not) and household income (greater than or equal to $90,000 or below; the approximate median of our sample) were also entered as indicator variables into all of our models. For these analyses ($N = 339$), we list-wise excluded 21 participants who did not report household income: 13 controls, 2 marrow donors, 1 directed kidney donor, 1 humanitarian aid worker, 2 non-directed kidney donors, 1 heroic rescuer, and 1 liver donor.

**Bootstrap sampling.** We next tested whether our finding that Honesty-Humility is the dimension of the HEXACO that most reliably distinguishes real-world extraordinary altruists would replicate in an independent control dataset. To accomplish this, we acquired data from a large population of 347,192 participants who completed the same HEXACO items measured in the present study. These participants were recruited through an online survey site (https://hexaco. org) from October 19, 2014 to October 18, 2018 (see Lee & Ashton, 2020). Although participants from Lee & Ashton completed the 100-item version of the HEXACO, we only used their scores computed from the same 60 items in the HEXACO-PI-R. We stratified the international dataset by country (United States), age (quantile split), and sex, and randomly drew 5,000 bootstrap samples without replacement that were matched to the full altruistic sample on each of these demographic variables. Because participants in the Lee & Ashton dataset were heavily skewed younger, we opted for a conservative bootstrap sample size ($N = 50$) to account for potential bias in participant selection during bootstrapping (i.e., we sought to mitigate potential bias for older participants closer to the age of our altruist sample being selected in majority of bootstrap samples). We then compared the altruist sample ($N = 347$) and our initial control sample ($N = 207$) to this new distribution of 5,000 mean scores for each of the HEXACO personality dimensions. Statistical inference was conducted by comparing altruist and control mean scores to the distribution of bootstrap sample means. We calculated the $p$-value as the proportion of instances in which bootstrap sample means exceeded the mean score for the comparison group. This analysis was carried out using custom code in Python 3.7.6 with the following packages: scipy (version 1.4.1)[86], numpy (version 1.18.1)[87], and pandas (version 1.0.1)[88].

### Study 1b

**Participants.** We invited participants from our original sample pool to partake in a second online study, during which we administered a social discounting task[19,42,43]. A sample of 291 adult participants (184 female), ages 19–75, opted into this study. The sample included 229 altruists and 62 controls. Altruists included the following: bone marrow donors ($N = 42$), directed kidney donors ($N = 58$), humanitarian aid workers ($N = 43$), non-directed kidney donors ($N = 59$), heroic rescuers ($N = 15$), liver donors ($N = 12$; of these, directed, $N = 9$; non-directed, $N = 3$).

In total 16 participants were excluded for choice inconsistencies in the social discounting task (i.e. switching between sharing and keeping

amounts of money more than two times within one or more task blocks). These participants included 4 controls, 6 marrow donors, 1 directed kidney donor, 2 humanitarian aid workers, 1 non-directed kidney donor, 1 heroic rescuer, and 1 liver donor (directed). Thus, 275 participants were included in the analytic sample (see Supplementary Information, Table S9). The participants excluded from social discounting analyses were included in all other analyses if they had complete data for the measures used in those analyses. In other words, all usable data were included in all analyses.

**Social discounting.** Participants first read the following standard instructions[42,43]: "Imagine that you have made a list of the 100 people closest to you in the world… The person at number one would be someone you know well and is your closest friend or relative. The person at #100 might be someone you recognize and encounter but perhaps you may not even know their name." Then, seven randomly presented blocks inquiring about sharing resources with person number 1, 2, 5, 10, 20, 50, or 100 on the list followed. Within a block, each trial presented 9 binary choices to keep or forgo a certain amount of money for the specified person, for example: "Please indicate which option you prefer: A) Keep $155 for myself alone. B) Keep $75 for myself and share $75 with person #10 on the list."

Values in the task ranged from $155 to $75 in decreasing $10 increments, which allowed for estimation of an "indifference point" when the respondent switched from selfish choices to sharing with a given person number on the list. The indifference point was calculated for each social other ($N$) by solving for the value at which the probability ($P$) between choosing a selfish versus generous option was 50% via binomial logistic regression:

$$P(share)_{trials} = \frac{\exp(b_0 + b_1 * Amount_{trials})}{1 + \exp(b_0 + b_1 * Amount_{trials})} \quad (2)$$

$$\frac{P}{1-P} = \exp(b_0 + b_1 * Amount_{trials}) \quad (3)$$

$$\ln\left(\frac{P}{1-P}\right) = \ln\left(\frac{.5}{1-.5}\right) = b_0 + b_1 * Amount_{indifference\ trial} \quad (3)$$

$$\ln(1) = 0 = b_0 + b_1 * Amount_{indifference\ trial} \quad (4)$$

$$-\frac{b_0}{b_1} = Amount_{indifference\ trial} \quad (5)$$

If the selfish option was chosen for all trials in the block, the indifference point was assumed at $70, and if the generous option was chosen for all trials in the block the indifference point was assumed at $160. Amounts willing to forgo ($v$) were calculated by subtracting $75 from the indifference point. Thus, we had seven "amount willing to forgo" ($v$) observations corresponding to one of seven social others ($N$) for every participant ($i$).

**Mixed-effects modeling.** Social distance ($N$) was centered at $N = 1$ (decision towards closest other); and groups were coded as indicator variables with typical adults entered as the baseline group. We used the nlme package version 3.1–145 (https://cran.r-project.org/package=nlme)[89] in R version 3.6.3 to fit data to a hyperbolic mixed-effects model using maximum likelihood estimation (MLE), which allowed both intercepts ($v_{0i}$) and discounting rates ($logk_i$) to vary across participants with group and covariates entered at level 2. Because discounting rates are non-parametrically distributed, we employed a variation in the hyperbolic discounting function to improve model convergence, estimating $logk$ rather than $k$[90]:

$$v = \frac{v_{0i}}{1 + k_i * N_i} = \frac{v_{0i}}{1 + \exp(logk_i) * N_i} \quad (6)$$

The model assessed how discounting rates ($logk_i$) differed across altruistic groups relative to controls controlling for the individual-level covariates age and sex:

Level 1 (hyperbolic function):

$$v_{Ni} = \frac{v_{0i}}{1 + \exp(logk_i) * N_i} + e_{Ni} \quad (7)$$

Level 2:

$$v_{0i} = \beta_{00} + r_{0i} \quad (8)$$

$$\begin{aligned} logk_i = \beta_{10} &+ \beta_{11} * LiverDonor_i + \\ &\beta_{12} * HeroicRescuer_i + \\ &\beta_{13} * NDKidneyDonor_i + \\ &\beta_{14} * DKidneyDonor_i + \\ &\beta_{15} * AidWorker_i + \\ &\beta_{16} * MarrowDonor_i + \\ &\beta_{17} * Age_i + \\ &\beta_{18} * Sex_i + r_{1i} \end{aligned} \quad (9)$$

To validate the estimated $logk$ parameters across subjects, we correlated them with a model-agnostic measure of social discounting: area-under-the-curve (AUC), which does not assume hyperbolicity in responding[91]. AUC was calculated for each participant by normalizing amount willing to forgo ($v$) as a percentage of maximum $v$, normalizing social distance ($N$) as a percentage of maximum $N$, connecting the crossover points by straight lines, then summing the areas of the trapezoids formed. As generosity increases, $logk$ decreases and AUC increases (both representing shallower slopes). We observed a significant negative correlation (Pearson-$r = -0.94$), with $logk$ capturing more variation across participants (see Supplementary Information, Figure S8), validating our use of $logk$ to index discounting rates.

In a step-wise manner, we added honesty-humility and the Group×Honesty-Humility interaction at level 2 into the selected model with group indicators and covariates.

We also conducted a series of linear ordinary least squares regression models to assess how each group differed from typical adults in amount willing to forgo for each of the seven social others. In each model, we entered group as an indicator variable with controls as the baseline group and controlled for age and sex.

Post-hoc analyses also included education and household income as covariates. Education (completed four-year degree or not) and household income (greater than or equal to $90,000 or below; the approximate median of our sample) were also entered as indicator variables into all of our models. For these analyses ($N = 264$), we listwise excluded 11 participants who did not report household income: 5 controls, 1 marrow donor, 1 heroic rescuer, 1 humanitarian aid worker, 1 directed kidney donor, 1 non-directed kidney donor, and 1 liver donor.

**Classifying altruists versus controls.** We next performed a penalized classification analysis aimed at accounting for any variance shared across all characteristics in order to predict the probability that a respondent is an altruist (all groups together) versus control. We created a binary outcome variable in which altruists were coded as 1 and controls as 0. Using the caret package version 6.0–86 (https://cran.r-project.org/package=caret) in R version 3.6.3, we randomly

partitioned data from our sub-sample of 275 participants into a training set (220 participants; 80%) and testing set (55 participants; 20%). With our training partition, we then trained a penalized logistic classifier with L1 regularization (LASSO) and 5-fold cross-validation to classify altruists versus controls using 25 predictor variables. These variables included: social discounting rate ($logk$, estimated from a mixed-effects model without level 2 group indicators or covariates), the six HEXACO dimensions, the four subscales of the IRI (empathic concern, personal distress, perspective taking, and fantasy), the five risk taking and five risk perception subscales of the DOSPERT (social, ethical, financial, health/safety, and recreational), and the three major subscales of psychopathy (self-centered impulsivity, coldheartedness, and fearless dominance). Our selected model was the simplest model within one standard deviation of the optimal $\lambda$ parameter, which shrinks coefficients of less contributive variables toward zero. The model selected five variables: social discounting rate ($logk$), honesty-humility, openness to experience, social risk-taking, personal distress. We assessed performance by testing the classifier on the independent testing data (unseen by classifier) and calculating the area under the receiver-operating-characteristics curve, which reduces potential bias due to unbalanced classes. Lastly, we assessed how strongly these six selected variables, controlling for age, sex, education, and household income, predicted the probability of being an altruist using all observations with available data ($N = 264$; see Supplementary Information Table S12) using a binary logistic model.

### Study 2a
All study 2a procedures were carried out in accordance with a protocol approved by the Institutional Review Board at Linfield College in McMinnville, Oregon, and all participants provided electronic written informed consent upon enrollment.

**Participants.** A total of 319 participants were recruited by Qualtrics from a representative panel of potential U.S. participants. Panel demographics were matched with the U.S. population based on the following criteria: sex (51% Female), age (30.5% 18–34 years, 34.4% 35–54 years, 35.2% 55+ years), race/ethnicity (62.3% non-Hispanic white, 12.4% non-Hispanic black, 17.3% Hispanic, 5.4% Asian, 2.6% other), and education (24% HS degree or less, 48% Bachelors or some college, 28% graduate or professional degree). We recruited this representative sample while concurrently accounting for participants that that failed to pass an attention check that explicitly asked them to choose specific responses. We excluded 111 participants based on this criterion. The remaining 208 participants were included for analysis and approximately corresponded to target demographics in terms of sex (50% female, 48.1% male, 1.9% other), age (30.3% 18–34 years, 34.1% 35–54 years, 35.6% 55+ years), race/ethnicity (67.3% non-Hispanic white, 13% non-Hispanic black, 11.5% Hispanic, 5.3% Asian, 2.9% other), and education (21.6% HS degree or less, 54.8% Bachelors or some college, 23.6% graduate or professional degree).

**Procedure.** To assess perceptions of altruists and their respective prosocial behaviors, we asked participants to rate various individuals in terms of nine dimensions: altruism, risk, the six HEXACO dimensions, and social discounting. Participants provided their perceptions of the average person, of exemplars of each of the types of six types of altruists that were the focus of our study (liver donors, heroic rescuers, non-directed kidney donors, humanitarian aid workers, directed kidney donors, and bone marrow donors), and six groups who are notable for non-altruistic actions included to minimize participants' ability to discern the focus of the study (internet trolls, foster parents, tax evaders, mountaineers, blood donors, and marathon runners). For each dimension, groups were displayed in random order and rated using 5-point scales. Definitions of each trait and descriptions of each group were provided (see Supplementary Information, Table S16a, b).

Participants also completed a third-person social discounting task, which assessed judgments about how an individual who had engaged in each of the six forms of altruism (as well as an average person) would be likely to allocate resources for others. Participants were instructed to imagine that [an individual representing each group] made a list of 100 people closest to him or her in the world ranging from their closest friend or relative at #1 to a mere acquaintance or stranger at #100. They were then instructed to move a slider bar "to indicate how much of their resources this person would sacrifice for each of the following people." They were presented with seven slider bars (for person #1, #2, #5, #10, #20, #50, and #100) which could be moved from 0 (sacrifice no resources; keep all for self) to 100 (sacrifice all resources; give all away). These ratings directly represented judgments about the amount the target would be willing to forgo ($v_{beliefs}$). Social discounting rates ($logk_{beliefs}$) were estimated using the hyperbolic discounting model separately for each group for each participant.

**Statistical analyses.** To assess how participants rated each type of altruist in comparison to the average person on each characteristic, we employed a series of linear mixed-effects regressions using the lme function within the nlme package version 3.1-145 in R version 3.6.3, which allowed intercepts to vary across participants. In each model, we entered groups as indicator variables with "average person" as the baseline group.

### Study 2b (Pre-registered)
To confirm our findings, we pre-registered (pre-registration link: https://osf.io/7t4qf/) on 2022-07-12 at 1:06 PM and conducted a second study 2b. Procedures were carried out in accordance with a protocol approved by the Institutional Review Board at Linfield College in McMinnville, Oregon, and all participants provided electronic written informed consent upon enrollment. We did not deviate from procedures outlined in the pre-registration.

**Participants.** We again aimed to recruit a sample of 200 American adults using a Qualtrics panel designed to match census-based U.S. population demographics: sex (51% Female), age (30.5% 18–34 years, 34.4% 35–54 years, 35.2% 55+ years), race/ethnicity (62.3% non-Hispanic white, 12.4% non-Hispanic black, 17.3% Hispanic, 5.4% Asian, 2.6% other), and education (24% HS degree or less, 48% Bachelors or some college, 28% graduate or professional degree). We recruited this representative sample while concurrently accounting for participants that that failed to pass the inclusion criteria we explicitly outlined in the pre-registration. A total of 201 participants were recruited by Qualtrics from a representative panel of potential U.S. participants approximately corresponded to target demographics in terms of sex (50.25% female, 50.25% male, 0.5% other), age (30.85% 18–34 years, 33.83% 35–54 years, 35.32% 55+ years), race/ethnicity (60.2% non-Hispanic white, 12.44% non-Hispanic black, 16.92% Hispanic or Latino, 4.98% Asian, 5.47% other), and education (23.88% HS degree or less, 47.76% Bachelors or some college, 28.36% graduate or professional degree).

**Procedure.** Our procedures followed those outlined in Study 2a and our pre-registration. To assess perceptions of altruists and their respective prosocial behaviors, we asked an independent sample of participants to rate various individuals in terms of nine dimensions: altruism, risk, the six HEXACO dimensions, and social discounting. Participants provided their perceptions of the average person, of exemplars of each of the types of six types of altruists that were the focus of our study (liver donors, heroic rescuers, non-directed kidney donors, humanitarian aid workers, directed kidney donors, and bone marrow donors), and six groups of who are notable for non-altruistic actions included to minimize participants' ability to discern the focus

of the study (internet trolls, tax evaders, marathon runners, analog astronauts, former contestants in the national Miss America pageant, and BASE jumpers). For each dimension, groups were displayed in random order and rated using 5-point scales. Definitions of each trait and descriptions of each group were provided (see Supplementary Information, Table S16a, b).

Participants also completed a third-person social discounting task, which assessed judgments about how an individual who had engaged in each of the six forms of altruism (as well as an average person) would be likely to allocate resources for others. Participants were instructed to imagine that [an individual representing each group] made a list of 100 people closest to him or her in the world ranging from their closest friend or relative at #1 to a mere acquaintance or stranger at #100. They were then instructed to move a slider bar "to indicate how much of their resources this person would sacrifice for each of the following people." They were presented with seven slider bars (for person #1, #2, #5, #10, #20, #50, and #100) which could be moved from 0 (sacrifice no resources; keep all for self) to 100 (sacrifice all resources; give all away). These ratings directly represented judgments about the amount the target would be willing to forgo ($v_{beliefs}$). Social discounting rates ($logk_{beliefs}$) were estimated using the hyperbolic discounting model separately for each group for each participant.

**Statistical analyses.** To assess how participants rated each type of altruist in comparison to the average person on each characteristic, we employed a series of linear mixed-effects regressions using the lme function within the nlme package version 3.1-145 in R version 3.6.3, which allowed intercepts to vary across participants. In each model, we entered groups as indicator variables with "average person" as the baseline group.

## Study 2c

To explore whether this prediction error on behalf of the general population was specific to altruism, or whether it could be due to their rarity, we conducted a study in which we recruited two independent samples from rare populations. All study 2c procedures were carried out in accordance with a protocol approved by the Institutional Review Board at Georgetown University, Washington DC, and all participants provided electronic written informed consent upon enrollment.

**Participants.** We recruited two independent samples of rare populations ($N = 28$), including extreme athletes (e.g., BASE Jumpers) and former contestants in the national Miss America pageant. Recruitment of BASE Jumpers was carried out using previous recruitment pools from researchers who have assessed personality traits of these athletes[48–51] as well as online public forums (e.g., https://www.blincmagazine.com/forum/). Recruitment of Miss America contestants was carried out via snowball sampling and via online forums (e.g., Facebook groups).

**Bootstrap sampling.** To generate demographically-matched control samples for each of these new rare groups, we used the same bootstrapping procedure described above. We again used data from a large population of 347,192 participants who completed the same HEXACO items measured in the present study. These participants were recruited through an online survey site (https://hexaco.org) from October 19, 2014 to October 18, 2018 (see Lee & Ashton, 2020). Although participants from Lee & Ashton completed the 100-item version of the HEXACO, we only used their scores computed from the same 60 items in the HEXACO-PI-R. We stratified the international dataset by country (United States), age (quantile split), and sex (female only for Miss America contestants), and randomly drew 5000 bootstrap samples without replacement that were matched on each of these

demographic variables to the Miss America contestant and BASE Jumper samples, respectively. Because participants in the Lee & Ashton dataset were heavily skewed younger, we again opted for the same conservative bootstrap sample size ($N = 50$) to account for potential bias in participant selection during bootstrapping (i.e., we sought to mitigate potential bias for older participants closer to the age of our rare samples being selected in majority of bootstrap samples). We then compared the Miss America contestant and BASE Jumper samples to their respective bootstrap distribution of 5000 control mean scores for each of the HEXACO personality dimensions. Statistical inference was conducted by comparing Miss America contestant and BASE Jumper mean scores to their respective distribution of bootstrap sample control means. We calculated the $p$-value as the proportion of instances in which bootstrap sample means exceeded the mean score for the comparison group.

**Statistical analyses.** We then tested whether the general population from the second sample of participants ($N = 201$; Study 2b) would uniquely link each group to these characteristics. To assess how participants in the general population rated each rare group in comparison to the average person on each characteristic, we employed a series of linear mixed-effects regressions using the lme function within the nlme package version 3.1-145 in R version 3.6.3, which allowed intercepts to vary across participants. In each model, we entered groups as indicator variables with "average person" as the baseline group.

**Representational similarity analysis.** Due to the nature of our data, we did not have a direct approach of mapping common perceptions onto actual characteristics using first-order parametric statistics. Therefore, we employed a method from systems and cognitive neuroscience that utilizes second-order statistics called representational similarity analysis (RSA)[53] using the lsan_tools package (https://github.com/LabSocialAffectNeuro/lsan_tools)[92] in Python 3.7.6 to test how perceptions map onto actual characteristics of six different, real-world populations of altruists. We selected actual characteristics of the six HEXACO personality traits and their corresponding perceptual ratings for each of studies 2a and 2b. To construct a similarity space of actual altruism across trait dimensions, we first computed standardized scores for each group by dividing the group mean by its standard error, then $Z$-scored across groups within each trait dimension. Then, we computed the pairwise absolute difference among all groups and their characteristics, resulting in a 36 × 36 true representational dissimilarity matrix (RDM). Likewise, to construct a similarity space of altruism perceptions for each of the two samples, we again computed standardized scores for each of the group ratings by dividing the mean by its standard error, then $Z$-scored across groups within each trait dimension. Then, we computed the pairwise absolute difference among all group characteristic ratings, again resulting in a 36 × 36 perceptual RDM for each characteristic. Each matrix consisted of 630 unique combinations of group characteristic pairs (36! / 2!×(36-2)!), with larger values corresponding to greater dissimilarity between pairs.

We assessed pairwise correspondence between the lower triangles of each pair of RDMs using the Spearman-rank correlation. Statistical significance was assessed at $\alpha = .05$ using a Mantel permutation test[93] by computing the same Spearman *rho* statistic after shuffling the group labels 5,000 times to generate an empirical null distribution. We calculated the $p$-value as the proportion of instances in which the permuted statistic exceeded the true Spearman *rho* statistic.

## Reporting summary

Further information on research design is available in the Nature Portfolio Reporting Summary linked to this article.

## Data availability

The processed data generated in this study are available on the Open Science Framework [https://osf.io/7t4qf/]. The raw data are protected and are not available due to data privacy laws. The bootstrap samples in study 1 were generated from data used in Lee & Ashton (2020) and Lee & Ashton (2018) with permission from the authors. These data can be requested from the authors.

## Code availability

Jupyter Notebooks for code in R/Python are shared on the Open Science Framework at https://osf.io/7t4qf/.

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

## Acknowledgements
This study was supported by the Kuno Award for Applied Science for the Social Good (S&R Evermay) and National Science Foundation (NSF) grant (#1729406) to A.A.M. We would like to thank Dan Dickinson (Transplant Village), Laura Vincent (National Marrow Donation Program), Merry Duffy (Be The Match®), Rob Williams (International Medical Corps and Médicins Sans Frontieres/Doctors Without Borders), Eric Zahren (Carnegie Hero Fund Commission), Dr. Omer Mei-Dan, Dr. Allison Porter, Professor Wendy Mendes, and Montana L. Ploe, whose recruitment efforts made this study possible. We also thank Rilee Macaluso, Isabelle Way, Jenna Hessel, and Angelica Gomez Horta for their help with study 2, Kathleen Neill for her help aggregating per capita frequency rate and demographic data of each altruistic group, Rebecca M. Ryan for her consultation on statistical methods. We also wish to express our gratitude to the participants who contributed their time and energy to this work.

## Author contributions
A.A.M. conceptualized study. K.M.V., K.O., and S.A.R. designed the study 1ab procedure. M.K.W., A.A.M., and S.A.R. designed and collected data for study 2a. S.A.R. and A.A.M. designed and collected data for study 2bc. H.S.E. coordinated communications with partners for study recruitment and data collection for study 1. D.G.R. assisted with recruitment. S.A.R. analyzed the data and wrote the manuscript with A.A.M. All co-authors revised and approved the manuscript.

## Competing interests
The authors declare no competing interests.
