## [Peer Review File · Nature Communications]

Unselfish traits and social decision-making patterns characterize six populations of real-world extraordinary altruistsReviewers' Comments:

Reviewer #1:

Remarks to the Author:

This is a review of the manuscript „Unselfish traits and social decision-making patterns characterize six populations of real-world extraordinary altruists“. I genuinely enjoyed reading the manuscript. I find the topic of investigating the personality profiles of 'extraordinary altruists' very interesting (in my opinion, also for readers of Nature Communications), the recruited altruist-samples impressive, the studies/analyses thoroughly described, and the manuscript well written overall. Frankly speaking, I do not have any major concerns, as I do believe that the recruitment of the altruist-samples clearly provides a strong contribution to the literature on selfish traits. That being said, I have a few comments that might be helpful for further improving the manuscript.

- As much as I was excited about the altruist-samples, I was a little bit disappointed about the samples forming the control group as well as the 'prediction' group. With regard to the control group, I would have expected both a larger sample as well as some attempts to match participants with participants from the altruist-group (although it is briefly described why no matching was attempted, I think it would have been beneficial). With regard to the 'prediction' group, I would have expected a second group formed by 'experts' (i.e., people working in these areas such as doctors performing kidney donations, journalists interviewing heroic rescuers, researchers on unselfish traits). Given that this cannot be changed now, I would recommend placing more emphasis on the first part (comparing 'extraordinary altruists' with the control group), as compared to the prediction. In this regard, it might also be possible to compare the Ms and SDs of the measures in the altruist-groups to the Ms and SDs of these measures reported in the literature based on larger 'normal population' samples. Yes, this is also not a perfect comparison, but it might mitigate the concern that the comparison group was biased in a way.

- The HEXACO model explicitly contains an interstitial scale among Honesty-Humility, Emotionality, and Agreeableness, termed Altruism. Although this scale has hardly been used in research (in fact, I am also not a big fan of it myself), I found it a bit surprising that neither this nor any other altruism-scale (e.g., Rushton, 1980) was administered. Maybe this could be discussed.

- To potentially facilitate the bigger picture, one could consider moving an analysis with comparing only one overall altruist-group to the control group to the main text/tables.

- Even though I find the altruism-sample/s impressive, I would still not consider the investigation as 'large-scale'. While this is correct in relative terms, it certainly is not in absolute/normative terms, given that many trait-studies are based on several thousands of participants nowadays.

- Readers not very familiar with trait research might wonder about why traits seem to play a particular role in novel situations or under extraordinary circumstances. Maybe this could be explained in a little bit more detail.

Again, thank you very much for the opportunity to review this manuscript.

Reviewer #2:

Remarks to the Author:

This paper examines which psychological traits and decision-making patterns consistently distinguish six types of extremely altruistic actors (e.g., living kidney donors, humanitarian workers, etc) from control samples. Moreover, the authors examine whether a typical, diverse American sample can predict these differences.

I appreciate and applaud many features of the paper, including the exceptional samples, careful measurement, and clear writing. This was truly a pleasure to read. I agree with the authors claim that this work meaningfully extends upon past research by Vekaria et al. (2017) by investigating the link between social discounting and exceptional altruism in various extra-ordinarily altruistic samples. I also find it fascinating and important that the authors observe a great deal of similarity between the six extraordinary altruist samples here. I encourage the authors to discuss this similarity, and its potential meaning, in their paper.

Alongside these important strengths, I also had a number of questions and concerns that tempered my enthusiasm.

1. I did not see any mention of the present study design, sample size, analytic plan, exclusion decisions, etc being pre-registered. Like most research, this work appears to include a number of "researcher degrees of freedom" that can drastically inflate type-I error and lead to false conclusions (Simmons, Nelson & Simonsohn, 2011). Such decisions may seem trivial in isolation, but there seem to be many small decisions that could sway conclusions (e.g., controlling for age and gender in all analyses; excluding 16 participants with choice inconsistencies in Study 1b). While I appreciate that access to such exceptional samples is incredibly time consuming and difficult, I think this underscores the importance of pre-determining and committing to one's key decisions so that conclusions from this rare sample provide the most robust evidence possible. Along similar lines, it wasn't clear to me whether the authors corrected their alpha rates for multiple comparisons. My sincere apologies if I missed details regarding pre-registration and planned analyses in the manuscript. I saw a link to data on the OSF, but I couldn't find a pre-registration link.

2. The authors explain that they did not seek a control sample to match the extraordinary altruist sample on key dimensions. As I understand it, the authors did not do so because some types of extraordinary altruists are selected to match certain demographics, but I'm not sure how or why that precludes this effort. I would encourage the authors to consider recruiting a matched control sample on key dimensions (e.g., age, gender, education, income, religiosity) as a more stringent test.

3. The central findings of this paper compare extraordinary altruists to the control samples in Studies 1a/b. The extraordinary altruist sample is identified by their previous behavior and rare actions, but who are the control sample? In particular, I'd be curious to know if/how the control sample matches the general population on the key dimensions of interests: HEXACO dimensions, social discounting, personal distress, risk-taking, the IRI, etc. This information is critical to report in text because the present results could just as likely be due to reports of a particularly selfish Control group (rather than a particularly caring/unselfish group of Altruists).

4. To what extent are the authors concerned that their Extraordinary Altruist sample reports are inflated by the fact that not all Exceptional Altruists agreed to participate? This may have biased their sample to the kindest of the kind (the cream of the already outstanding crop).

5. Were Extraordinary Altruists aware that they had been recruited/invited to participate because of their past behavior? If so, it seems like this could raise concerns of demand characteristics or biased responding wherein the Altruists inflate their care or concern for others to appear consistent with their exceptional past acts.

6. The authors include Study 2 to examine whether the general population is able to predict the dimensions that distinguish Extraordinary Altruists from others. The general population sample used here is 107 people recruited on Qualtrics, which is not representative or large. More importantly the authors note that the general population sample "did not perceive that altruists are distinguished by unselfish traits (Honesty-Humility) or behavior (social-discounting) or any of the other traits assessed." It seems that the authors are interpreting a null result as evidence. A null result cannot be

proven with Null Hypothesis Testing. I would be more convinced here if the authors conducted Bayesian analyses to show overwhelming evidence of the null, and/or if the authors had an extremely large and representative with statistical power to detect a small effect but then found no evidence for difference.

7. One reason that the general population sample in Study 2 may have been unable to predict the traits or behavior of extraordinary altruists is because of the rarity of these actions. People have very little exposure and insight into these infrequent acts. Is it possible to distinguish the findings in Study 2 from a generalized prediction error that emerges when people don't have much information of a target? The authors note that participants were asked to predict the traits and behaviors of some other, non-altruistic targets (e.g., marathon runners, tax evader, internet troll). Were participants any more/less accurate in predicting the traits and behaviors of these semi-rare groups? Note: Even here, I'm guessing the base rates of marathon runners and internet trolls is higher than the extraordinary altruist acts, and so this is an imperfect solution.

Reviewer #3:

Remarks to the Author:

The present research investigated several individual differences variables that best distinguish extraordinary altruists from typical adults. High Honesty-Humility, low Social Distancing, and low Personal Distress were found to figure more prominently in distinguishing the two groups than did other individual differences variables such as empathy, boldness, and various risk taking measures. The present research is remarkable in a number of ways: Although there have been some studies involving real-world extraordinary prosocial behaviors in the past, the present research stands out from those previous studies in terms of the number and types of real-world extraordinary altruists included in the study. In addition, the research questions answered by these remarkable data are critical ones that have not been fully recognized by researchers in social and personality psychology. I am sure that the present research will be one of my all-time favorite empirical studies in psychology.

Let me first begin with the significance of the present findings in relation to Honesty-Humility (HH). As the authors correctly described (see also Diebels, Leary, & Chon, 2018), the primary content of HEXACO-PI-R Honesty-Humility involves a lack of selfishness (such as low materialism, low self-entitlement, low exploitiveness, etc.) rather than prosociality per se. Drawing on this observation, some personality psychologists predict that HH will primarily predict antisocial behaviors rather than prosocial behaviors. Ashton and Lee (in press; Lee & Ashton, in press) tried to address this widely shared misperception (e.g., point # 12 in Ashton & Lee, in press) by citing some works showing the importance of "selfishness" in predicting active prosocial behaviors using some economic games and a small-scale unpublished study involving kidney donors. The present findings could have been decisively important in addressing this point, given its large sample of real-world extraordinary altruists as well as the inclusion of diverse variables beyond the HEXACO variables.

I do not have many substantive or methodological suggestions to provide, but I do have a couple of suggestions which may help readers to better understand the results of the present research.

I would've wanted to see the means and standard deviations of all the psychological variables for each of the six altruists groups and the control group. I believe the authors should provide tables for this information. From this information, readers should have a better sense of how "large" those coefficients are in the regression analysis tables. Second, future researchers may want to compare scores on the psychological variables of real-life extraordinary altruists against other groups of their interest. In that regard, it seems to me that the control group participants in the present research generally score quite higher in "moral characters" than do typical adults (i.e., it is possible that they are more socialized morally). For example, from Table S3a, I learn that female means for the HEXACO-60 scales are (in the HEXACO order) 3.78, 3.24, 3.49, 3.22, 3.89, and 3.74. The

corresponding numbers from the HEXACO online sample (N = 48562 women, with age distribution quite similar to that of the control group) are 3.48, 3.33, 3.24, 2.96, 3.63, and 3.74 (recomputed for the HEXACO-60 scales from Lee & Ashton, 2018). Thus, the control group means are about 0.5 SD unit higher than those of the online sample for the Honesty-Humility and Conscientiousness scales. Therefore, the authors should make a prominent mention that differences between the extraordinary altruists and typical adults may have been substantially underestimated for some psychological variables in the present research.

Line 252, "Generally, discounting rates decreased as Honesty-Humility increased (Figure 3, Table 1), consistent with increased Honesty-Humility supporting reduced social discounting rates among altruists." I was curious about the strength of the relationship between HH and social discounting rate. I was wondering if a more straightforward measure such as correlation coefficient can be reported. Come to think of it, it should be useful to report the correlation matrix involving all the psychological variables in a supplementary table.

Line 372, I personally found the finding of Study 2 very interesting too; i.e., people generally do not have intuition about the rather strong positive link between HH traits/social discounting rate to extraordinary altruism. I am aware the results of a conference study in which a researcher assessed HEXACO personality traits of kidney donors (Maynard, 2019, N = approximately 60). This study used the HEXACO-100 so it was possible to look into the facet-level differences. Consistent with the findings from the present research, the best three facet-level predictors all come from the Honesty-Humility domain: Fairness, Greed-avoidance (low materialism), and Modesty. Even experienced researchers would not have been able to predict that (a lack of) exploitiveness, materialism, and self-entitlement can best characterize real-world extraordinary altruists more strongly than, or at least as strongly as, overall Altruism [sympathy, generosity, warm-heartedness] does. In this sense, the results of the present research provide a significant insight into human prosociality.

Line 187, "conducted a" was repeated.

Ashton, M. C., & Lee, K. (in press). Objections to the HEXACO model of personality structure—And why those objections fail. *European Journal of Personality*.

Diebels, K. J., Leary, M. R., & Chon, D. (2018). Individual differences in selfishness as a major dimension of personality: A reinterpretation of the sixth personality factor. *Review of General Psychology*, 22(4), 367-376.

Lee, K., & Ashton, M. C. (2018). Psychometric properties of the HEXACO-100. *Assessment*, 25(5), 543-556.

Maynard, C. (2019, April). Take us off your pedestal: Normalizing unspecified kidney donation. ELPAT Conference, Krakow, Poland.

Reviewer #4:

Remarks to the Author:

General comments

The authors find that a sample of extraordinary altruists (relative to controls) score higher on self-reported humility/honest, lower on personal distress, and exhibit shallower social discounting in a hypothetical decision-making task.

The primary challenge is that the data are conditioned people having *already* engaged in acts of extraordinary altruism. I have difficulty believing that this constellation of traits/preferences is

discriminantly predictive of extraordinary altruism. Sure, once I've given a kidney or rescued someone from a burning building, I see myself as someone who values others over myself (and report as such on personality scales). The problem is that there are likely MANY more people who score high on H, low on PD, and have shallow social discounting rates than there are people giving up kidneys or running into burning buildings (though I'd be unsurprised to learn that they're all nurses, public school teachers, and social workers). So what is actually doing the work in the moment that people make these kinds of extraordinary decisions? Or as the authors ask in the first paragraph "what generally distinguishes individuals who engage in stringently defined real-world altruism is unselfishness?" We can't answer this question with these data.

Specific comments

"A widespread assumption across the social and biological sciences and among the general public is that selfish motivations underlie all human behavior, including prosocial behaviors¹⁻⁴." With all due respect, this is a strawman. Social and biological sciences are not universally, nor are all their theories Hobbesian. In fact several of the authors on this paper have made this point very persuasively over the last decade if not longer! This framing mischaracterizes the literature and undermines the paper.

p. 5 typo: "We conducted a conducted a series of..."

"Because our groups differed in average gender distribution and age " Why not compare them to age- and gender-matched controls? Surely it would be easy to administer these batteries (e.g., online) to a large control panel and then match from there.

"For every unit increase in Honesty-Humility score, participants were 83.47% more likely to be altruists" Is this a typo? This is one of those findings that would make me go back and double check the analyses. If this were true we'd expect to have much shorter kidney waitlists than we currently have.

"However, they did not perceive that altruists are distinguished by unselfish traits (Honesty-Humility) or behavior (social discounting)" For what it is worth I wouldn't have thought honesty-humility is associated with being unselfish either because those words mean different things. (In fact reading this paper inspired me to go look at the H items and my conclusion is that Honesty/Humility is a poor label for that factor; it's really more like egalitarianism or universalism.) As for the social distancing null, this concerns me greatly because if anything this item should have acted a *comprehension check* in these data. Participants didn't think a person who sacrificed their safety/well-being/comfort to help a stranger would also be more likely to give up money for a stranger than a control??

"These findings position social discounting patterns as a meaningful, replicable, non-self-report-based index of individual variation in ecologically valid altruism." I don't see how one can draw this conclusion. For example, conservatives are MORE compassionate and giving to those close to them relative to liberals, whereas liberals have flatter curves and are more generous to those farther from them (Waytz et al., 2019; Enke et al., 2019) but how can you determine which is more altruistic? Wouldn't it make more sense to fix the target and then administer the Social Value Orientation measure to determine who is more altruistic at each level?

Thank you very much for considering our manuscript. We appreciate the reviewers' encouraging comments and feedback. We also thank the editor for the opportunity to appeal your initial decision by responding to the reviewers' critiques and submitting a revised manuscript that responds to these suggestions.

Reviewer #1 (Remarks to the Author):

This is a review of the manuscript "Unselfish traits and social decision-making patterns characterize six populations of real-world extraordinary altruists". I genuinely enjoyed reading the manuscript. I find the topic of investigating the personality profiles of 'extraordinary altruists' very interesting (in my opinion, also for readers of Nature Communications), the recruited altruist-samples impressive, the studies/analyses thoroughly described, and the manuscript well written overall. Frankly speaking, I do not have any major concerns, as I do believe that the recruitment of the altruist-samples clearly provides a strong contribution to the literature on selfish traits. That being said, I have a few comments that might be helpful for further improving the manuscript.

We thank the reviewer for these encouraging comments, for taking the time to review our manuscript, and for offering the suggestions below to improve our manuscript.

- As much as I was excited about the altruist-samples, I was a little bit disappointed about the samples forming the control group as well as the 'prediction' group. With regard to the control group, I would have expected both a larger sample as well as some attempts to match participants with participants from the altruist-group (although it is briefly described why no matching was attempted, I think it would have been beneficial).

We apologize that our wording in the Limitations section was misleading regarding the important issue of matching altruists and controls. We recognize that our explanation in the original manuscript had some shortcomings [we wrote: "these populations (particularly organ and marrow donors) normally are selected to donate based on measures related to age and health. Because of this, and because altruist groups vary widely in their demographic makeup, we did not limit control participant selection by matching them to each group on demographics"]. This excerpt was meant to highlight (as we describe in our responses to Reviewers 2 and 4) that each altruistic population varies in its demographics. Therefore, we could not match each of the altruistic groups to each other and could also not match each group to a single control group.

We have now revised this section to clarify that we did seek to match our control participants to the *combined* altruist sample. Our recruitment strategies explicitly sought controls of similar approximate demographics as altruists (including similar gender, race/ethnicity, age, and education). However, exact matching could not be achieved due to our concurrent recruitment of altruists and controls (meaning we did not know precisely how old, educated, etc. altruists would be in advance). This recruitment strategy achieved approximate demographic matching in gender (60.81% female altruists in comparison to 64.73% female controls; $\chi^2(1)=.85$, $p=.356$) and household income (see Table S2 for percentage breakdown; $\chi^2(8)=14.93$, $p=.061$). However, altruists were older (mean=44.06 years, SD=12.50) in contrast to controls (mean=37.71 years, SD=9.07), $T(552)=6.37$, $p<.001$) and less educated (73.49% altruists received a college

education in contrast to 87.92% controls, $\chi^2(1)=16.22$, $p<.001$). What group differences emerged despite our efforts were statistically controlled for in all analyses.

We appreciate that we should have made these issues clearer, and have now done so in our Limitations section:

It should also be noted that altruists (particularly organ and marrow donors) are often selected based on factors related to age, sex, and health. **In part because of this, altruist groups vary widely in their demographic makeup (see Supplementary Information, Table S1). This demographic variation could be considered an additional reason that the consistency observed in altruists' personality traits across the six groups is noteworthy and compelling. It is also a reason we did not recruit controls to match each individual group demographically, but rather to match the combined altruist sample. Our recruitment strategies explicitly sought controls of similar approximate demographics as altruists, and achieved approximate demographic matching in gender and income (see Supplementary Information, Table S2) although exact matching of age and education was not achieved during concurrent recruitment. What group differences emerged despite our efforts were statistically controlled for in all analyses. (pg. 10)**

We also added text to our Results section to make this clearer:

We conducted a series of multiple linear regression analyses to assess how altruists were distinguished from controls across these measures (including subscale scores). **Controls were recruited to approximately match altruist demographics. This recruitment strategy achieved approximate demographic matching in gender (60.81% female altruists in comparison to 64.73% female controls; $\chi^2(1)=.85$, $p=.356$) and household income (see Supplementary Information, Table S2, for percentage breakdown; $\chi^2(8)=14.93$, $p=.061$). However, because groups were recruited concurrently, altruists were older (mean=44.06 years, $SD=12.50$) in contrast to controls (mean=37.71 years, $SD=9.07$), $T(552)=6.37$, $p<.001$) and less educated (73.49% altruists received a college education in contrast to 87.92% controls, $\chi^2(1)=16.22$, $p<.001$) (see Supplementary Information, Table S2). Therefore, all regressions controlled for age and gender (among all participants), as well as income and education (among participants with available data), which varied across altruistic groups... We conducted follow-up analyses that also controlled for age, gender, education, and household income (for all participants with available data; $N=533$) and that yielded similar results (see Supplementary Information, Tables S3a-1). (pg. 5)**

With regard to the ‘prediction’ group, I would have expected a second group formed by ‘experts’ (i.e., people working in these areas such as doctors performing kidney donations, journalists interviewing heroic rescuers, researchers on unselfish traits). Given that this cannot be changed now, I would recommend placing more emphasis on the first part (comparing ‘extraordinary altruists’ with the control group), as compared to the prediction. In this regard, it might also be possible to compare the Ms and SDs of the measures in the altruist-groups to the Ms and SDs of these measures reported in the literature based on larger ‘normal population’ samples. Yes, this is also not a perfect

comparison, but it might mitigate the concern that the comparison group was biased in a way.

We appreciate this suggestion, and we have now put more emphasis on our result regarding the altruist versus control comparisons throughout our manuscript.

It should be noted that because we recruited controls in an effort to match them demographically with altruists—we agree with the reviewer about the importance of this goal—we would not expect them to exactly match ‘normal population’ samples. In terms of how they might compare to such samples, we are grateful to Reviewer 3 for their valuable insights on this topic. Reviewer 3 writes:

In that regard, it seems to me that the control group participants in the present research generally score quite higher in “moral characters” than do typical adults (i.e., it is possible that they are more socialized morally). For example, from Table S3a, I learn that female means for the HEXACO-60 scales are (in the HEXACO order) 3.78, 3.24, 3.49, 3.22, 3.89, and 3.74. The corresponding numbers from the HEXACO online sample (N = 48562 women, with age distribution quite similar to that of the control group) are 3.48, 3.33, 3.24, 2.96, 3.63, and 3.74 (recomputed for the HEXACO-60 scales from Lee & Ashton, 2018). Thus, the control group means are about 0.5 SD unit higher than those of the online sample for the Honesty-Humility and Conscientiousness scales. Therefore, the authors should make a prominent mention that differences between the extraordinary altruists and typical adults may have been substantially underestimated for some psychological variables in the present research.

In keeping with this suggestion, we now include in our Discussion that our reported effect sizes may be underestimates—although this is speculative because there is no population level study reporting population means and standard deviations for the HEXACO-60 (these scores were calculated using a subset of answers to the HEXACO-100 in Lee & Ashton, 2018), and that we have been unable to find representative population-level studies of IRI, DOSPERT, PPI, social discounting, etc.:

These results are particularly notable considering that people who are higher in prosociality are more likely to participate in psychological research to begin with⁵¹—thus the observed differences between altruists and controls may be underestimates (i.e., population-level differences may be larger). (pg. 9)

- The HEXACO model explicitly contains an interstitial scale among Honesty-Humility, Emotionality, and Agreeableness, termed Altruism. Although this scale has hardly been used in research (in fact, I am also not a big fan of it myself), I found it a bit surprising that neither this nor any other altruism-scale (e.g., Rushton, 1980) was administered. Maybe this could be discussed.

We appreciate the reviewer making this point! The interstitial "Altruism" scale comprised of 4 question items is unfortunately only available in the 100-item HEXACO personality inventory, which we did not use (we used the 60-item HEXACO Personality Inventory-Revised). However,

we agree that this measure of altruism is flawed, as are many other self-reported altruism scales. In our previous work, we have used the Rushton scale and the Social Value Orientation scale and found they did not correspond at all to extraordinary altruism (Brethel-Haurwitz et al., 2016). This underscores a main motivation for studying populations of real-world altruists, such as those included in the present study, and a main reason that we believe our findings represent a significant advance—because they indicate that measures of selfishness predict real-world altruism better than some direct measures of altruism do. We have added this to the discussion section:

The measures collected in the present study do not represent an exhaustive list of measures potentially related to real-world altruism. Although we did not pre-register these measures, they were selected to be as comprehensive as possible (the six-factor inventory) and to capture variables consistently linked to altruism in prior research, including Agreeableness⁷, Honesty-Humility^{19,45}, risk sensitivity⁶⁸, empathy^{10,11}, social discounting¹⁸, and psychopathy¹⁸... In prior work, we have found that extraordinary altruists are not distinguished from controls on measures of self-reported altruism⁶⁷, social value orientation, or altruistic punishment²¹. (pg. 11)

- To potentially facilitate the bigger picture, one could consider moving an analysis with comparing only one overall altruist-group to the control group to the main text/tables.

We agree with this idea, and now report means and standard deviations of our psychological variables for comparison between the combined altruist sample compared to the control sample in Figures S1a-e and Table S7.

- Even though I find the altruism-sample/s impressive, I would still not consider the investigation as ‘large-scale’. While this is correct in relative terms, it certainly is not in absolute/normative terms, given that many trait-studies are based on several thousands of participants nowadays.

We have now removed phrasing of large-scale that might imply large-scale in an absolute sense and instead focus on the relative sense of this being the largest investigation of extraordinary altruists to date.

- Readers not very familiar with trait research might wonder about why traits seem to play a particular role in novel situations or under extraordinary circumstances. Maybe this could be explained in a little bit more detail.

Again, thank you very much for the opportunity to review this manuscript.

Thank you for pointing this out and for your helpful suggestions. We have updated the following text in the Introduction section regarding why traits seem to play a particular role in novel situations or under extraordinary circumstances:

Several lines of evidence point to extraordinary altruism reflecting stable unselfish traits. Stable trait-level tendencies that bias ecological and social behaviors in consistent ways

are observed across organisms of many species¹³⁻¹⁶ and are most predictive of behavioral outcomes in novel contexts or those lacking strong norms¹⁷. **In such situations, no information about how to behave adaptively is known, such that situational constraints on behavior are weaker and individual differences are most likely to be revealed¹⁷.** The rarity of extraordinary acts like heroic rescues and altruistic organ donations renders contexts in which these acts occur novel and lacking strong norms by default, and thus particularly likely to correspond to dispositional variation. (pg. 3)

Reviewer #2 (Remarks to the Author):

This paper examines which psychological traits and decision-making patterns consistently distinguish six types of extremely altruistic actors (e.g., living kidney donors, humanitarian workers, etc) from control samples. Moreover, the authors examine whether a typical, diverse American sample can predict these differences.

I appreciate and applaud many features of the paper, including the exceptional samples, careful measurement, and clear writing. This was truly a pleasure to read. I agree with the authors claim that this work meaningfully extends upon past research by Vekaria et al. (2017) by investigating the link between social discounting and exceptional altruism in various extra-ordinarily altruistic samples.

Thank for these encouraging comments, for taking the time to review our study, and providing us with helpful suggestions to improve the manuscript.

I also find it fascinating and important that the authors observe a great deal of similarity between the six extraordinary altruist samples here. I encourage the authors to discuss this similarity, and its potential meaning, in their paper.

We agree that the similarities observed among altruistic groups are interesting. Particularly, given the notable differences in the demographics of the 6 groups of extraordinary altruists and the differences in the forms of altruism they have engaged in (from acts of physical heroism to the decision to donate bone marrow), we found it compelling that a common set of traits persistently distinguished them from controls, and that all three of these traits were associated with unselfishness. We appreciate the encouragement to emphasize this point in our manuscript.

We have added the following excerpt to our Discussion section:

That extraordinary altruists are consistently distinguished by a common set of traits linked to unselfishness is particularly compelling given the differences in the demographics of the various altruistic groups and the differences in the forms of altruism they have engaged in—from acts of physical heroism to the decision to donate bone marrow. This finding replicates and extends findings from a previous study¹⁸ demonstrating that extraordinary altruists show heightened subjective valuation of socially distant others. (pg. 9)

Alongside these important strengths, I also had a number of questions and concerns that tempered my enthusiasm.

1. I did not see any mention of the present study design, sample size, analytic plan, exclusion decisions, etc being pre-registered. Like most research, this work appears to include a number of “researcher degrees of freedom” that can drastically inflate type-I error and lead to false conclusions (Simmons, Nelson & Simonsohn, 2011). Such decisions may seem trivial in isolation, but there seem to be many small decisions that could sway conclusions (e.g., controlling for age and gender in all analyses; excluding 16 participants

with choice inconsistencies in Study 1b). While I appreciate that access to such exceptional samples is incredibly time consuming and difficult, I think this underscores the importance of pre-determining and committing to one's key decisions so that conclusions from this rare sample provide the most robust evidence possible. Along similar lines, it wasn't clear to me whether the authors corrected their alpha rates for multiple comparisons. My sincere apologies if I missed details regarding pre-registration and planned analyses in the manuscript. I saw a link to data on the OSF, but I couldn't find a pre-registration link.

We share the reviewer's sentiment regarding the importance of pre-registering study designs and analytic plans for replicability and for the sake of transparency, a practice our lab now employs regularly. We did not pre-register this particular study, which was conceptualized and designed in 2017, when preregistration was still relatively uncommon (prevalence at journals like Psychological Science grew from 2% in 2016 to 30% in 2019; see <https://www.psychologicalscience.org/publications/observer/obsonline/easier-done-than-said-lessons-from-6-years-of-preregistration.html>).

Despite this, we sought to mitigate spurious correlations by taking a within-lab multiverse analytic approach (Stegen et al., 2016), which refers to conducting analyses of interest across the whole set of data sets that arise from different reasonable choices for data processing. We analyzed the data using multiple strategies for the sake of openness, replicability, transparency, and robustness of findings (and will make all data and code available on the Open Science Framework (OSF) for reproducibility). For example, we first compared the altruistic groups to controls via multiple regression controlling for age and gender in our full sample, then controlling for age, gender, income, and education in a sample with available data (Tables S3a-1). Third, we assessed which factors account for the most variance in altruist versus controls controlling for age, gender, income, and education (Table 2). All of these analyses revealed very similar results. We have made this issue clear in our Limitations section.

The measures collected in the present study do not represent an exhaustive list of measures potentially related to real-world altruism. Although we did not pre-register these measures, they were selected to be as comprehensive as possible (the six-factor inventory) and to capture variables consistently linked to altruism in prior research, including Agreeableness⁷, Honesty-Humility^{19,45}, risk sensitivity⁶⁸, empathy^{10,11}, social discounting¹⁸, and psychopathy¹⁸. We analyzed the data using multiple strategies for the sake of openness, replicability, transparency, and robustness of findings (and all data and code are available on the Open Science Framework for reproducibility). (pg. 11)

For all of our statistical analyses (linear and nonlinear) that compared altruistic groups to controls, we "dummy coded" the groups, which allowed us to simultaneously test how each altruistic group differed from the control group (which served as the implicit baseline group) within a single model. In other words, in each model the regression coefficient for each altruistic group represents the mean difference between that altruistic group versus the control group. Because these tests were non-independent and we did not conduct any subsequent independent post-hoc tests, we did not further correct for multiple comparisons.

2. The authors explain that they did not seek a control sample to match the extraordinary altruist sample on key dimensions. As I understand it, the authors did not do so because some types of extraordinary altruists are selected to match certain demographics, but I'm not sure how or why that precludes this effort. I would encourage the authors to consider recruiting a matched control sample on key dimensions (e.g., age, gender, education, income, religiosity) as a more stringent test.

As we describe above in response to Reviewer 1, we apologize that our wording in the Limitations section was misleading regarding the important issue of matching altruists and controls. We recognize that our explanation in the original manuscript had some shortcomings [we wrote: "these populations (particularly organ and marrow donors) normally are selected to donate based on measures related to age and health. Because of this, and because altruist groups vary widely in their demographic makeup, we did not limit control participant selection by matching them to each group on demographics"]. This excerpt was meant to highlight (as we describe in our responses to Reviewer 1 and 4) that each altruistic population varies in its demographics. Therefore, we could not match each of the altruistic groups to each other and could also not match each group to a single control group.

We have now revised this section to clarify that we did seek to match our control participants to the *combined* altruist sample. Our recruitment strategies explicitly sought controls of similar approximate demographics as altruists (including similar gender, race/ethnicity, age, and education). However, exact matching could not be achieved due to our concurrent recruitment of altruists and controls (meaning we did not know precisely how old, educated, etc. altruists would be in advance). This recruitment strategy achieved approximate demographic matching in gender (60.81% female altruists in comparison to 64.73% female controls; $\chi^2(1)=.85$, $p=.356$) and household income (see Table S2 for percentage breakdown; $\chi^2(8)=14.93$, $p=.061$). However, altruists were older (mean=44.06 years, $SD=12.50$) in contrast to controls (mean=37.71 years, $SD=9.07$), $T(552)=6.37$, $p<.001$) and less educated (73.49% altruists received a college education in contrast to 87.92% controls, $\chi^2(1)=16.22$, $p<.001$). What group differences emerged despite our efforts were statistically controlled for in all analyses.

We have reservations about collecting new data for a control group during the current global pandemic, recession, and protracted period of social isolation and political unrest. (All data reported in Study 1 were collected prior to the pandemic.) Our own research (Vieira et al., 2020) and that of other groups indicates these ongoing issues affect a number of social and affective outcomes related to altruism, emotion, and personality (Sutin et al., 2020; Van de Groep et al., 2020) which could create confounds. A major advantage of our current control sample is that because they were concurrently recruited with altruists, there are no potentially hidden effects of the pandemic, recession, etc. that could be difficult to account for.

We appreciate that we should have made these issues clearer, and have now done so in our Limitations section:

It should also be noted that altruists (particularly organ and marrow donors) are often selected based on factors related to age, sex, and health. **In part because of this, altruist groups vary widely in their demographic makeup (see Supplementary Information, Table**

S1). This demographic variation could be considered an additional reason that the consistency observed in altruists' personality traits across the six groups is noteworthy and compelling. It is also a reason we did not recruit controls to match each individual group demographically, but rather to match the combined altruist sample. Our recruitment strategies explicitly sought controls of similar approximate demographics as altruists, and achieved approximate demographic matching in gender and income (see Supplemental Information, Table S2) although exact matching of age and education was not achieved during concurrent recruitment. What group differences emerged despite our efforts were statistically controlled for in all analyses. (pg. 10)

We also added text to our Results section to make this clearer:

We conducted a series of multiple linear regression analyses to assess how altruists were distinguished from controls across these measures (including subscale scores). Controls were recruited to approximately match altruist demographics. This recruitment strategy achieved approximate demographic matching in gender (60.81% female altruists in comparison to 64.73% female controls; $\chi^2(1)=.85$, $p=.356$) and household income (see Supplementary Information, Table S2, for percentage breakdown; $\chi^2(8)=14.93$, $p=.061$). However, because groups were recruited concurrently, altruists were older (mean=44.06 years, $SD=12.50$) in contrast to controls (mean=37.71 years, $SD=9.07$), $T(552)=6.37$, $p<.001$) and less educated (73.49% altruists received a college education in contrast to 87.92% controls, $\chi^2(1)=16.22$, $p<.001$) (see Supplementary Information, Table S2). Therefore, all regressions controlled for age and gender (among all participants), as well as income and education (among participants with available data), which varied across altruistic groups... We conducted follow-up analyses that also controlled for age, gender, education, and household income (for all participants with available data; $N=533$) and that yielded similar results (see Supplementary Information, Tables S3a-1). (pg. 5)

3. The central findings of this paper compare extraordinary altruists to the control samples in Studies 1a/b. The extraordinary altruist sample is identified by their previous behavior and rare actions, but who are the control sample? In particular, I'd be curious to know if/how the control sample matches the general population on the key dimensions of interests: HEXACO dimensions, social discounting, personal distress, risk-taking, the IRI, etc. This information is critical to report in text because the present results could just as likely be due to reports of a particularly selfish Control group (rather than a particularly caring/unselfish group of Altruists).

It should be noted that because we recruited controls in an effort to match them demographically with altruists—we agree with the reviewer about the importance of this goal—we would not expect them to exactly match ‘normal population’ samples. In terms of how they might compare to such samples, we are grateful to Reviewer 3 for their valuable insights on this topic. Reviewer 3 writes:

In that regard, it seems to me that the control group participants in the present research generally score quite higher in “moral characters” than do typical adults (i.e., it is possible that they are more socialized morally). For example, from Table S3a, I learn that

female means for the HEXACO-60 scales are (in the HEXACO order) 3.78, 3.24, 3.49, 3.22, 3.89, and 3.74. The corresponding numbers from the HEXACO online sample (N = 48562 women, with age distribution quite similar to that of the control group) are 3.48, 3.33, 3.24, 2.96, 3.63, and 3.74 (recomputed for the HEXACO-60 scales from Lee & Ashton, 2018). Thus, the control group means are about 0.5 SD unit higher than those of the online sample for the Honesty-Humility and Conscientiousness scales. Therefore, the authors should make a prominent mention that differences between the extraordinary altruists and typical adults may have been substantially underestimated for some psychological variables in the present research.

In keeping with this suggestion, we now include in our Discussion that our reported effect sizes may be underestimates—although this is speculative because there is no population level study reporting population means and standard deviations for the HEXACO-60 (these scores were calculated using a subset of answers to the HEXACO-100 in Lee & Ashton, 2018), and that we have been unable to find representative population-level studies of IRI, DOSPRT, PPI, social discounting, etc.:

These results are particularly notable considering that people who are higher in prosociality are more likely to participate in psychological research to begin with⁵¹—thus the observed differences between altruists and controls may be underestimates (i.e., population-level differences may be larger). (pg. 9)

4. To what extent are the authors concerned that their Extraordinary Altruist sample reports are inflated by the fact that not all Exceptional Altruists agreed to participate? This may have biased their sample to the kindest of the kind (the cream of the already outstanding crop).

This is always a concern with not only extraordinary altruists, but in any research sample in which participation is voluntary—including controls in this study (see comments above). We have added this point to our Discussion section, as we believe it is worth considering that all human subjects research likely overestimates prosociality in typical samples because these samples are necessarily composed of those who volunteer to participate (Van Lange et al., 2011). We believe this issue is probably mitigated in extraordinary altruist samples composed of those who go to extraordinary lengths to help others, because participating in a short research study is such a small ask by comparison. Anecdotally, our experience has been that extraordinary altruists are more willing to participate in research than controls, which would artificially decrease rather than inflate our effect sizes, consistent with the suggestions of Reviewer 3.

These results are particularly notable considering that people who are higher in prosociality are more likely to participate in psychological research to begin with⁵¹—thus the observed differences between altruists and controls may be underestimates (i.e., population-level differences may be larger). (pg. 9)

5. Were Extraordinary Altruists aware that they had been recruited/invited to participate because of their past behavior? If so, it seems like this could raise concerns of demand

characteristics or biased responding wherein the Altruists inflate their care or concern for others to appear consistent with their exceptional past acts.

Our research is modeled on the approaches used in clinical and other special population research. All clinical research and research on special populations necessarily requires targeting the specific participants you are recruiting, whether they be participants who have depression, stroke, psychopathy, are within a certain age range, etc. We are experienced in conducting clinical and special populations research and endeavor in our recruitment, consenting, and instructional material to not be specific about our research goals to reduce demand characteristics. We say, for example, that we are studying social perceptions and behaviors, and provide a wide array of questions and instruments (as described) so that our specific hypotheses will not be apparent. Supporting the effectiveness of this approach, extraordinary altruists in our past studies tend not to score high on self-report altruism scales (Brethel-Haurwitz et al., 2016; Vekaria et al., 2020) and tend to report in interviews that they are not especially altruistic or unusual in any way, but simply had the right information at the right time (Marsh, 2017)—responses consistent with their high Honesty-Humility scores but not with the hypothesis that their responses primarily reflect demand.

6. The authors include Study 2 to examine whether the general population is able to predict the dimensions that distinguish Extraordinary Altruists from others. The general population sample used here is 107 people recruited on Qualtrics, which is not representative or large. More importantly the authors note that the general population sample “did not perceive that altruists are distinguished by unselfish traits (Honesty-Humility) or behavior (social-discounting) or any of the other traits assessed.” It seems that the authors are interpreting a null result as evidence. A null result cannot be proven with Null Hypothesis Testing. I would be more convinced here if the authors conducted Bayesian analyses to show overwhelming evidence of the null, and/or if the authors had an extremely large and representative with statistical power to detect a small effect but then found no evidence for difference.

In response to this and other reviewers' concerns, we have now collected a new, larger sample of participants recruited by Qualtrics to match US Census population characteristics in terms of gender (50% female, 48.1% male, 1.9% other), age (30.3% 18-34 years, 34.1% 35-54 years, 35.6% 55-65 years), race/ethnicity (67.3% non-Hispanic white, 13% non-Hispanic black, 11.5% Hispanic, 5.3% Asian, 2.9% other), and education (21.6% HS degree or less, 54.8% Bachelors or some college, 23.6% graduate or professional degree) and that is twice the size of our original sample (N=208). We found that this representative sample of American adults did not specifically associate extraordinary altruism with traits related to unselfishness, but rather with undifferentiated traits broadly construed as positive but not closely linked to actual altruism, including high Extraversion, high Agreeableness, and high Conscientiousness. We hope the reviewer agrees that these results are more compelling, as they are not null findings, but they nonetheless do not reflect a close match between perceptions and actual traits of extraordinary altruists. We now report:

Linear mixed-effects modeling revealed that respondents perceive all forms of extraordinary altruism to be both relatively altruistic and risky (see Supplementary

Information, Table S11). They also judged altruists as differing from the average person in terms of all six HEXACO personality dimensions. Estimated social discounting rates corresponding to judgments about how much an individual is willing to forgo for various social others ($\log k_{beliefs}$) were shallower for all altruistic groups relative to those estimated about the average person. Thus, respondents correctly viewed altruists as more willing to forgo resources for others at varying social distances (social discounting)—linking altruism to unselfish behavior. However, they did not specifically associate extraordinary altruism with traits related to unselfishness. Rather, they associated extraordinary altruism with undifferentiated traits broadly construed as positive. (see Supplementary Information, Figure S3ab). (pg. 8)

In addition, in response to Reviewer 1, we now put less emphasis on our Study 2 results throughout our manuscript.

7. One reason that the general population sample in Study 2 may have been unable to predict the traits or behavior of extraordinary altruists is because of the rarity of these actions. People have very little exposure and insight into these infrequent acts. Is it possible to distinguish the findings in Study 2 from a generalized prediction error that emerges when people don't have much information of a target? The authors note that participants were asked to predict the traits and behaviors of some other, non-altruistic targets (e.g., marathon runners, tax evader, internet troll). Were participants any more/less accurate in predicting the traits and behaviors of these semi-rare groups? Note: Even here, I'm guessing the base rates of marathon runners and internet trolls is higher than the extraordinary altruist acts, and so this is an imperfect solution.

We agree that the rarity of extraordinary altruism is one reason it could be difficult to predict traits of extraordinary altruists, and now have added the following text to the Discussion section:

The rarity of these forms of altruism may have made it more difficult for our population sample to predict what traits would distinguish altruists. (pg. 10)

Unfortunately, because we do not have any equivalent data in populations of marathon runners, etc. we are unable to compare accuracy across these targets.

Reviewer #3 (Remarks to the Author):

The present research investigated several individual differences variables that best distinguish extraordinary altruists from typical adults. High Honesty-Humility, low Social Distancing, and low Personal Distress were found to figure more prominently in distinguishing the two groups than did other individual differences variables such as empathy, boldness, and various risk taking measures. The present research is remarkable in a number of ways: Although there have been some studies involving real-world extraordinary prosocial behaviors in the past, the present research stands out from those previous studies in terms of the number and types of real-world extraordinary altruists included in the study. In addition, the research questions answered by these remarkable data are critical ones that have not been fully recognized by researchers in social and personality psychology. I am sure that the present research will be one of my all-time favorite empirical studies in psychology.

Let me first begin with the significance of the present findings in relation to Honesty-Humility (HH). As the authors correctly described (see also Diebels, Leary, & Chon, 2018), the primary content of HEXACO-PI-R Honesty-Humility involves a lack of selfishness (such as low materialism, low self-entitlement, low exploitiveness, etc.) rather than prosociality per se. Drawing on this observation, some personality psychologists predict that HH will primarily predict antisocial behaviors rather than prosocial behaviors. Ashton and Lee (in press; Lee & Ashton, in press) tried to address this widely shared misperception (e.g., point # 12 in Ashton & Lee, in press) by citing some works showing the importance of “selfishness” in predicting active prosocial behaviors using some economic games and a small-scale unpublished study involving kidney donors. The present findings could have been decisively important in addressing this point, given its large sample of real-world extraordinary altruists as well as the inclusion of diverse variables beyond the HEXACO variables.

We are grateful to the reviewer for these encouraging comments, for taking the time to review our study, and providing us with helpful suggestions to improve the manuscript. We were not aware of the Ashton & Lee (2020) manuscript pointing out how the fact that Honesty-Humility involves a lack of selfishness *should* mean it predicts prosociality, despite others' misperceptions it would not, and totally agree about the importance of selfishness (and its absence) to understanding prosociality. We agree that our data support this argument and have added these citations to support our arguments in the Discussion section.

Coupled with findings that low levels of unselfish traits (e.g., low Honesty-Humility, high social discounting) correspond to exploitative and antisocial behaviors such as cheating and aggression^{32,46}, these results also lend support to the notion of a bipolar caring continuum along which individuals vary in the degree to which they subjectively value (care about) the welfare of others⁴⁷⁻⁴⁹. They further suggest altruism—arguably the willingness to be voluntarily “exploited” by others—to be the opposite of phenotypes like psychopathy that are characterized by exploiting others⁵⁰. (pg. 9)

I do not have many substantive or methodological suggestions to provide, but I do have a couple of suggestions which may help readers to better understand the results of the present research.

I would've wanted to see the means and standard deviations of all the psychological variables for each of the six altruists groups and the control group. I believe the authors should provide tables for this information. From this information, readers should have a better sense of how "large" those coefficients are in the regression analysis tables.

We agree with this idea, and now report means and standard deviations of our psychological variables for comparison between the combined altruist sample (and each of the six altruist groups) compared to the control sample in Figures S1a-e and Table S7.

Second, future researchers may want to compare scores on the psychological variables of real-life extraordinary altruists against other groups of their interest. In that regard, it seems to me that the control group participants in the present research generally score quite higher in "moral characters" than do typical adults (i.e., it is possible that they are more socialized morally). For example, from Table S3a, I learn that female means for the HEXACO-60 scales are (in the HEXACO order) 3.78, 3.24, 3.49, 3.22, 3.89, and 3.74. The corresponding numbers from the HEXACO online sample (N = 48562 women, with age distribution quite similar to that of the control group) are 3.48, 3.33, 3.24, 2.96, 3.63, and 3.74 (recomputed for the HEXACO-60 scales from Lee & Ashton, 2018). Thus, the control group means are about 0.5 SD unit higher than those of the online sample for the Honesty-Humility and Conscientiousness scales. Therefore, the authors should make a prominent mention that differences between the extraordinary altruists and typical adults may have been substantially underestimated for some psychological variables in the present research.

This is very helpful insight. We now highlight that differences between extraordinary altruists and controls in our sample may have been underestimated, in light of estimates of population means.

These results are particularly notable considering that people who are higher in prosociality are more likely to participate in psychological research to begin with⁵¹—thus the observed differences between altruists and controls may be underestimates (i.e., population-level differences may be larger). (pg. 9)

Line 252, "Generally, discounting rates decreased as Honesty-Humility increased (Figure 3, Table 1), consistent with increased Honesty-Humility supporting reduced social discounting rates among altruists." I was curious about the strength of the relationship between HH and social discounting rate. I was wondering if a more straightforward measure such as correlation coefficient can be reported. Come to think of it, it should be useful to report the correlation matrix involving all the psychological variables in a supplementary table.

We have now added a correlation matrix reporting bivariate associations among all the psychological variables in the Supplementary Materials (Table S8). Honesty-Humility and social discounting rates (logk) were inversely correlated (*Spearman rho* = -.29).

Line 372, I personally found the finding of Study 2 very interesting too; i.e., people generally do not have intuition about the rather strong positive link between HH traits/social discounting rate to extraordinary altruism. I am aware the results of a conference study in which a researcher assessed HEXACO personality traits of kidney donors (Maynard, 2019, N = approximately 60). This study used the HEXACO-100 so it was possible to look into the facet-level differences. Consistent with the findings from the present research, the best three facet-level predictors all come from the Honesty-Humility domain: Fairness, Greed-avoidance (low materialism), and Modesty. Even experienced researchers would not have been able to predict that (a lack of) exploitiveness, materialism, and self-entitlement can best characterize real-world extraordinary altruists more strongly than, or at least as strongly as, overall Altruism [sympathy, generosity, warm-heartedness] does. In this sense, the results of the present research provide a significant insight into human prosociality.

We appreciate this comment! Cody Maynard is an altruistic kidney donor and is a colleague of our laboratory. We have spoken with him extensively about this project, and view the research he conducted as very interesting, if preliminary. We agree that our findings are in some senses counterintuitive, as they do not fit many common conceptualizations about the origins of altruism, and we share the reviewer's hope that the results of the present research will provide valuable insights.

Line 187, "conducted a" was repeated.

We have fixed this typo.

Ashton, M. C., & Lee, K. (in press). Objections to the HEXACO model of personality structure—And why those objections fail. *European Journal of Personality*.

Diebels, K. J., Leary, M. R., & Chon, D. (2018). Individual differences in selfishness as a major dimension of personality: A reinterpretation of the sixth personality factor. *Review of General Psychology*, 22(4), 367-376.

Lee, K., & Ashton, M. C. (2018). Psychometric properties of the HEXACO-100. *Assessment*, 25(5), 543-556.

Maynard, C. (2019, April). Take us off your pedestal: Normalizing unspecified kidney donation. ELPAT Conference, Krakow, Poland.

Reviewer #4 (Remarks to the Author):

General comments

The authors find that a sample of extraordinary altruists (relative to controls) score higher on self-reported humility/honest, lower on personal distress, and exhibit shallower social discounting in a hypothetical decision-making task.

We thank the reviewer for taking the time to review our manuscript and for offering the suggestions below to improve our manuscript.

The primary challenge is that the data are conditioned people having *already* engaged in acts of extraordinary altruism. I have difficulty believing that this constellation of traits/preferences is discriminantly predictive of extraordinary altruism. Sure, once I've given a kidney or rescued someone from a burning building, I see myself as someone who values others over myself (and report as such on personality scales). The problem is that there are likely MANY more people who score high on H, low on PD, and have shallow social discounting rates than there are people giving up kidneys or running into burning buildings (though I'd be unsurprised to learn that they're all nurses, public school teachers, and social workers). So what is actually doing the work in the moment that people make these kinds of extraordinary decisions? Or as the authors ask in the first paragraph "what generally distinguishes individuals who engage in stringently defined real-world altruism is unselfishness?" We can't answer this question with these data.

We appreciate the opportunity to clarify this issue, which we agree is important. The present study did not aim to ask "what is actually doing the work in the moment that people make these kinds of extraordinary decisions?" which we agree is difficult to determine for a variety of reasons. (Although our 2014 paper "Neural and cognitive characteristics of extraordinary altruists" did touch on this issue.)

We believe this is a different question than, "Is unselfishness what generally distinguishes individuals who engage in stringently defined real-world altruism?" We believe our data are equipped to make advances towards answering this question, and we would be the first attempt to do so across multiple populations of real-world altruists.

However, we share the reviewer's concerns of directionality (traits/behaviors predicting future real-world altruistic behavior) as we highlight in our Limitations section: "Temporal conclusions are also limited by the fact that altruistic behaviors were necessarily performed prior to testing." (pg 10). We have amended our text so that it is clear that whereas the measures we identify can distinguish individuals who have engaged in stringently defined real-world altruism, we do not show that we can prospectively predict these behaviors using these measures, and we appreciate the suggestion to make this distinction.

Based on our 10 years working with extraordinary altruists, we believe it unlikely that our findings reflect altruists concluding based on their past behavior that they are unusually altruistic or compassionate. Altruists do not score high on self-report altruism scales that are more

straightforwardly related to altruism (Brethel-Haurwitz et al., 2016; Vekaria et al., 2020) and in interviews they very consistently report that they are not especially altruistic or unusual in any way, but simply had the right information at the right time (Marsh, 2017)—a response consistent with the high Honesty-Humility scores observed in the present study. We now go into more detail on this topic in our Discussion.

Temporal conclusions are also limited by the fact that altruistic behaviors were necessarily performed prior to testing. **The measures we identify distinguish individuals who have engaged in stringently defined real-world altruism, rather than attempting to prospectively predict normatively rare altruistic behaviors. An alternate hypothesis could be that altruists' patterns of responding reflect their concluding based on their past behavior that they are unusually altruistic or compassionate. However, two lines of evidence argue against this possibility. First, altruists do not score highly on self-report altruism scales that are more straightforwardly related to altruism^{21,67} (including empathic concern in the present study) and in interviews they consistently report that they are not especially altruistic or unusual in any way, but simply had the right information at the right time^{18,68}—a response consistent with the high Honesty-Humility scores observed in the present study. Second, data from our population sample can be used as a proxy for the traits that the average person associates with extraordinary altruists. RSA found that these traits do not closely match the traits that altruists actually report, which suggests our altruists' traits do not likely simply reflect stereotypes about altruists that they possess.** (pg 10-11).

Specific comments

“A widespread assumption across the social and biological sciences and among the general public is that selfish motivations underlie all human behavior, including prosocial behaviors1–4.” With all due respect, this is a strawman. Social and biological sciences are not universally, nor are all their theories Hobbesian. In fact several of the authors on this paper have made this point very persuasively over the last decade if not longer! This framing mischaracterizes the literature and undermines the paper.

We thank the reviewer for this comment and appreciate this point. We have reduced emphasis on the assumption that selfish motivations underlie all human behavior throughout the manuscript.

p. 5 typo: “We conducted a conducted a series of...”

We thank the reviewer for spotting this typo and have corrected it.

“Because our groups differed in average gender distribution and age “ Why not compare them to age- and gender-matched controls? Surely it would be easy to administer these batteries (e.g., online) to a large control panel and then match from there.

As we describe above in response to Reviewers 1 and 2, we apologize that our wording in the Limitations section was misleading regarding the important issue of matching altruists and controls. We recognize that our explanation in the original manuscript had some shortcomings

[we wrote: "these populations (particularly organ and marrow donors) normally are selected to donate based on measures related to age and health. Because of this, and because altruist groups vary widely in their demographic makeup, we did not limit control participant selection by matching them to each group on demographics"]. This excerpt was meant to highlight (as we describe in our responses to Reviewer 1 and 2) that each altruistic population varies in its demographics. Therefore, we could not match each of the altruistic groups to each other and could also not match each group to a single control group.

We have now revised this section to clarify that we did seek to match our control participants to the *combined* altruist sample. Our recruitment strategies explicitly sought controls of similar approximate demographics as altruists (including similar gender, race/ethnicity, age, and education). However, exact matching could not be achieved due to our concurrent recruitment of altruists and controls (meaning we did not know precisely how old, educated, etc. altruists would be in advance). This recruitment strategy achieved approximate demographic matching in gender (60.81% female altruists in comparison to 64.73% female controls; $\chi^2(1)=.85$, $p=.356$) and household income (see Table S2 for percentage breakdown; $\chi^2(8)=14.93$, $p=.061$). However, altruists were older (mean=44.06 years, $SD=12.50$) in contrast to controls (mean=37.71 years, $SD=9.07$), $T(552)=6.37$, $p<.001$) and less educated (73.49% altruists received a college education in contrast to 87.92% controls, $\chi^2(1)=16.22$, $p<.001$). What group differences emerged despite our efforts were statistically controlled for in all analyses.

We have reservations about collecting new data for a control group during the current global pandemic, recession, and protracted period of social isolation and political unrest. (All data reported in Study 1 were collected prior to the pandemic.) Our own research (Vieira et al., 2020) and that of other groups indicates these ongoing issues affect a number of social and affective outcomes related to altruism, emotion, and personality (Sutin et al., 2020; Van de Groep et al., 2020) which could create potential confounds. A major advantage of our current control sample is that because they were concurrently recruited with altruists, there are no potentially hidden effects of the pandemic, recession, etc. that could be difficult to account for.

We appreciate that we should have made these issues clearer, and have now done so in our Limitations section:

It should also be noted that altruists (particularly organ and marrow donors) are often selected based on factors related to age, sex, and health. **In part because of this, altruist groups vary widely in their demographic makeup (see Supplementary Information, Table S1). This demographic variation could be considered an additional reason that the consistency observed in altruists' personality traits across the six groups is noteworthy and compelling. It is also a reason we did not recruit controls to match each individual group demographically, but rather to match the combined altruist sample. Our recruitment strategies explicitly sought controls of similar approximate demographics as altruists, and achieved approximate demographic matching in gender and income (see Supplemental Information, Table S2) although exact matching of age and education was not achieved during concurrent recruitment. What group differences emerged despite our efforts were statistically controlled for in all analyses. (pg. 10).**

We also added text to our Results section to make this clearer:

We conducted a series of multiple linear regression analyses to assess how altruists were distinguished from controls across these measures (including subscale scores). **Controls were recruited to approximately match altruist demographics. This recruitment strategy achieved approximate demographic matching in gender (60.81% female altruists in comparison to 64.73% female controls; $\chi^2(1)=.85$, $p=.356$) and household income (see Supplementary Information, Table S2, for percentage breakdown; $\chi^2(8)=14.93$, $p=.061$). However, because groups were recruited concurrently, altruists were older (mean=44.06 years, $SD=12.50$) in contrast to controls (mean=37.71 years, $SD=9.07$), $T(552)=6.37$, $p<.001$) and less educated (73.49% altruists received a college education in contrast to 87.92% controls, $\chi^2(1)=16.22$, $p<.001$) (see Supplementary Information, Table S2). Therefore, all regressions controlled for age and gender (among all participants), as well as income and education (among participants with available data), which varied across altruistic groups... We conducted follow-up analyses that also controlled for age, gender, education, and household income (for all participants with available data; $N=533$) and that yielded similar results (see Supplementary Information, Tables S3a-1). (pg. 5)**

“For every unit increase in Honesty-Humility score, participants were 83.47% more likely to be altruists” Is this a typo? This is one of those findings that would make me go back and double check the analyses. If this were true we’d expect to have much shorter kidney waitlists than we currently have.

We confirm that the value is accurate. It's important to note that because the base rate of these altruistic behaviors is so low (as we detail in our manuscript) even an 83.74% increase in that base rate still yields a low value. However, we understand how this framing can be confusing. We have edited the results to read (consistent with the odds ratios reported in Table 2):

For every unit increase in Honesty-Humility score, participants were **1.84 times** more likely to be altruists; for every unit increase in discounting rate, participants were **.20 times less** likely to be altruists; and for every unit increase in Personal Distress score, participants were **.09 times less** likely to be altruists. (pg. 7)

“However, they did not perceive that altruists are distinguished by unselfish traits (Honesty-Humility) or behavior (social discounting)” For what it is worth I wouldn’t have thought honesty-humilty is associated with being unselfish either because those words mean different things. (In fact reading this paper inspired me to go look at the H items and my conclusion is that Honesty/Humility is a poor label for that factor; it’s really more like egalitarianism or universalism.) As for the social distancing null, this concerns me greatly because if anything this item should have acted a *comprehension check* in these data. Participants didn’t think a person who sacrificed their safety/well-being/comfort to help a stranger would also be more likely to give up money for a stranger than a control??

In response to this and other reviewers' concerns, we have now collected a new, larger sample of participants recruited by Qualtrics to match US Census population characteristics in terms of gender (50% female, 48.1% male, 1.9% other), age (30.3% 18-34 years, 34.1% 35-54 years,

35.6% 55-65 years), race/ethnicity (67.3% non-Hispanic white, 13% non-Hispanic black, 11.5% Hispanic, 5.3% Asian, 2.9% other), and education (21.6% HS degree or less, 54.8% Bachelors or some college, 23.6% graduate or professional degree) and that is twice the size of our original sample (N=208). We found that this representative sample of American adults did not specifically associate extraordinary altruism with traits related to unselfishness, but rather with undifferentiated traits broadly construed as positive but not closely linked to actual altruism, including high Extraversion, high Agreeableness, and high Conscientiousness. We hope the reviewer agrees that these results are more compelling, as they are not null findings, but they nonetheless do not reflect a close match between perceptions and actual traits of extraordinary altruists. We now report:

Linear mixed-effects modeling revealed that respondents perceive all forms of extraordinary altruism to be both relatively altruistic and risky (see Supplementary Information, Table S11). They also judged altruists as differing from the average person in terms of all HEXACO personality dimensions. Estimated social discounting rates corresponding to judgments about how much an individual is willing to forgo for various social others ($\log k_{beliefs}$) were shallower for all altruistic groups relative to those estimated about the average person. Thus, respondents correctly viewed altruists as more willing to forgo resources for others at varying social distances (social discounting)—linking altruism to unselfish behavior. However, they did not specifically associate extraordinary altruism with traits related to unselfishness. Rather, they associated extraordinary altruism with undifferentiated traits broadly construed as positive. (see Supplementary Information, Figure S3ab). (pg. 8)

We are happy to confirm that, in line with the reviewer's prediction, changing our social discounting prediction task did in fact yield results that suggest this task is a comprehension check of sorts. However, participants nonetheless did not link altruism specifically to social discounting or Honesty-Humility (as we note in the manuscript we provided detailed definitions of each scale), and the results of our RSA analysis did not find close mapping between perceived and actual traits of altruists.

In addition, in response to Reviewer 1, we now put less emphasis on our results regarding these comparisons throughout our manuscript.

“These findings position social discounting patterns as a meaningful, replicable, non-self-report-based index of individual variation in ecologically valid altruism.” I don’t see how one can draw this conclusion. For example, conservatives are MORE compassionate and giving to those close to them relative to liberals, whereas liberals have flatter curves and are more generous to those farther from them (Waytz et al., 2019; Enke et al., 2019) but how can you determine which is more altruistic? Wouldn’t it make more sense to fix the target and then administer the Social Value Orientation measure to determine who is more altruistic at each level?

We thank the reviewer for this point and have now amended the sentence to read, "These findings position social discounting patterns as meaningful, replicable, non-self-report-based indices of individual variation in ecologically valid, extraordinary forms of altruism." (pg 9)

This change highlights the fact that our manuscript focuses on extraordinary altruism. We agree with the reviewer that sharing with both close and distant others are important forms of altruism. But making large sacrifices or taking significant risks for more distant others is inherently more extraordinary because it is much rarer across all populations (as indicated by the social discounting curve). This is a point we now make more explicitly in the Discussions as follows:

In that social discounting decisions directly reflect variation in the subjective value of outcomes for various others versus the self, they are valuable for understanding the basis of extraordinary altruism—**making large sacrifices or taking significant risks for more distant others is inherently more extraordinary because it is much rarer across all populations.** (pg. 9)

We did not include measures of Social Value Orientation (SVO) or a measure of altruistic punishment like that referenced in Enke (2019) because data from our previous research has indicated extraordinary altruists do not show meaningful differences from controls on either of these measures (Brethel-Haurwitz et al., 2016). We now highlighted this in our Discussion section:

The measures collected in the present study do not represent an exhaustive list of measures potentially related to real-world altruism. Although we did not pre-register these measures, they were selected to be as comprehensive as possible (the six-factor inventory) and to capture variables consistently linked to altruism in prior research, including Agreeableness⁷, Honesty-Humility^{19,45}, risk sensitivity⁶⁸, empathy^{10,11}, social discounting¹⁸, and psychopathy¹⁸... In prior work, we have found that extraordinary altruists are not distinguished from controls on measures of self-reported altruism⁶⁷, social value orientation, or altruistic punishment²¹. (pg. 11)

We did include the social discounting measure we describe here because our previous work has found that this paradigm yields substantial differences between extraordinary altruists and controls (Vekaria et al., 2017), a finding that was replicated here.

We are cautious about drawing parallels with the social discounting paradigm used in our study (in which participants allocate monetary resources towards specific people from their closest known other to an unknown stranger; a measure validated in a real-world sample of altruists, see Vekaria et al., 2017) and the moral allocation paradigm used in Waytz et al. (2019) (in which participants allocate abstract moral units towards different categories from "immediate family" to "all natural things in the universe including inert entities such as rocks") without direct evidence of the overlap in the constructs they assess. We recently conducted a pre-registered study (in preparation) that found no strong relationship between political ideology and social discounting with monetary allocations across the last four years, suggesting that the social discounting paradigm and moral allocation paradigm may measure dissociable outcomes.

We note that we do not claim that social discounting is the only possible meaningful, replicable, non-self-report-based index of individual variation in ecologically valid extraordinary altruism—although it was one of very few such measures in our large battery. The measures collected in the

present study do not represent an exhaustive list of measures potentially related to real-world altruism and agree that the Waytz moral allocation task would be a particularly interesting task to consider both in community samples and altruistic samples, and have highlighted this task as a potential future direction.

Other batteries of assessment should be administered among these altruistic populations to gain further insight into the characteristics related to extraordinary altruism, such as paradigms that map onto different types of prosocial decision-making according to divergent task and neural features⁷¹. These might include other economic games, those that examine moral decision-making^{56,72}, or those that investigate moral inferences about others⁷³. (pg. 11)

References

- Ashton, M. C., & Lee, K. (2020). Objections to the HEXACO Model of Personality Structure—And Why Those Objections Fail. *European Journal of Personality*, 34(4), 492–510. <https://doi.org/10.1002/per.2242>
- Brethel-Haurwitz, K. M., Stoycos, S. A., Cardinale, E. M., Huebner, B., & Marsh, A. A. (2016). Is costly punishment altruistic? Exploring rejection of unfair offers in the Ultimatum Game in real-world altruists. *Scientific Reports*, 6(1), 18974. <https://doi.org/10.1038/srep18974>
- Enke, B. (2019). Kinship, cooperation, and the evolution of moral systems. *The Quarterly Journal of Economics*, 134(2), 953–1019. <https://doi.org/10.1093/qje/qjz001>. Advance
- Lee, K., & Ashton, M. C. (2018). Psychometric Properties of the HEXACO-100. *Assessment*, 25(5), 543–556. <https://doi.org/10.1177/1073191116659134>
- Marsh, A. (2017). *The Fear Factor: How One Emotion Connects Altruists, Psychopaths, and Everyone In-Between*. Basic Books.
- Steegeen, S., Tuerlinckx, F., Gelman, A., & Vanpaemel, W. (2016). Increasing Transparency Through a Multiverse Analysis. *Perspectives on Psychological Science*, 11(5), 702–712. <https://doi.org/10.1177/1745691616658637>
- Sutin, A. R., Luchetti, M., Aschwanden, D., Lee, J. H., Sesker, A. A., Strickhouser, J. E., Stephan, Y., & Terracciano, A. (2020). Change in five-factor model personality traits during the acute phase of the coronavirus pandemic. *PLoS ONE*, 15(8), 1–13. <https://doi.org/10.1371/journal.pone.0237056>
- Van de Groep, S., Zanolie, K., Green, K. H., Sweijen, S. W., & Crone, E. A. (2020). A daily diary study on adolescents' mood, empathy, and prosocial behavior during the COVID-19 pandemic. *PLoS ONE*, 15(10), e0240349. <https://doi.org/10.1371/journal.pone.0240349>
- Van Lange, P. A. M., Schippers, M., & Balliet, D. (2011). Who volunteers in psychology experiments? An empirical review of prosocial motivation in volunteering. *Personality and Individual Differences*, 51(3), 279–284. <https://doi.org/10.1016/j.paid.2010.05.038>
- Vekaria, K. M., Brethel-Haurwitz, K. M., Cardinale, E. M., Stoycos, S. A., & Marsh, A. A. (2017). Social discounting and distance perceptions in costly altruism. *Nature Human Behaviour*, 1(5), 1–7. <https://doi.org/10.1038/s41562-017-0100>
- Vekaria, K. M., O'Connell, K., Rhoads, S. A., Brethel-Haurwitz, K. M., Cardinale, E. M., Robertson, E. L., Walitt, B., VanMeter, J. W., & Marsh, A. A. (2020). Activation in bed nucleus of the stria terminalis (BNST) corresponds to everyday helping. *Cortex*, 127, 67–77. <https://doi.org/10.1016/j.cortex.2020.02.001>
- Vieira, J., Pierzchajlo, S., Jangard, S., Marsh, A., & Olsson, A. (2020). Perceived threat and acute anxiety predict increased everyday altruism during the COVID-19 pandemic. *PsyArXiv*, 1–22. <https://doi.org/10.31234/osf.io/n3t5c>
- Waytz, A., Iyer, R., Young, L., Haidt, J., & Graham, J. (2019). Ideological differences in the expanse of the moral circle. *Nature Communications*, 10(1). <https://doi.org/10.1038/s41467-019-12227-0>

Reviewers' Comments:

Reviewer #1:

Remarks to the Author:

Thank you very much for the chance to review the revised version of the manuscript "Unselfish traits and social decision-making patterns characterize six populations of real-world extraordinary altruists". I think that the authors addressed most of the comments by (the other reviewers and) me in a convincing manner, including collecting new data and providing more information/results.

One comment that I still have is that I find the defense of the comparison sample not overly convincing—although I can see the arguments for not collecting data of a new comparison sample now. For instance, one could have aimed to recruit a larger comparison sample in order to ensure a proper matching; even the "matching on the group level" as argued by the authors has not worked out perfectly. I would thus recommend highlighting this limitation more clearly, especially because there are age and gender (but not educational level) effects for Honesty-Humility, based on the meta-analysis on the HEXACO-PI-R (Moshagen et al., 2019).

A little bit relatedly, the authors might want to be clearer about the limitation in terms of the temporal order of assessments. Theoretically, one could think about assessing characteristics in a large panel and then evaluating (e.g., after 10, 20 etc. years) who of the panel members has shown acts of extraordinary altruism (obviously, this comes with other limitations; but just to point out that the temporal order was not automatically given).

Another (new) comment that I have is that at the beginning and the end of the abstract, as well as at the beginning of the Introduction, the authors refer to the fact that extraordinary altruism is an evolutionary and motivational puzzle and that, e.g., "self-focused motivations [...] alone are insufficient explanations". The data, however, do in itself not provide new insights into the evolutionary or motivational reasons for such acts, which readers might expect to learn about. The study is good in describing differences between extraordinary altruists and other people, but the used measures (alone) do hardly help in shedding new light on the underlying (evolutionary or motivational) reasons. Maybe it would be good to adapt corresponding sentences, so that readers do not get wrong expectations and/or to be clear/er what the study adds (and what not). For instance, the first sentence of the Discussion does exactly this ("that generally characterizes them").

A smaller comment that I have is that previous research has aimed to link five-factor model (or Big Five) Agreeableness to altruism, whereas the present study linked HEXACO Agreeableness vs Anger to altruism. There are some noteworthy differences between FFM/Big Five and HEXACO Agreeableness (also depending on the FFM/Big Five measure used). Thus, proponents of the five-factor (vs. HEXACO) approach could argue that this study is not automatically in contrast to previous findings (looking at "not-extraordinary" altruism), but this is implied sometimes (e.g., in the Discussion "previously linked to altruism, including self-reported empathy, agreeableness..."). The authors might want to be more precise about this.

On a minor note, my previous comment concerning the interstitial altruism scale in the HEXACO-PI-R was not meant to imply that this measure is "flawed", but rather that I was surprised that this measure (or any other measure aiming to assess a construct termed "altruism") was not used. For instance, one could have just added the four (our eight) altruism-items from the HEXACO-PI-R to the HEXACO-60.

Smaller points:

- Maybe use "expect" than "recognize" on line 361?
- I don't understand what is meant by "the right information at the right time – a response consistent with the high Honesty-Humility scores" (line 451/452). HH scores do not say anything about which information people have (I think)?

Reviewer #2:

Remarks to the Author:

I appreciate the chance to review this manuscript again and I thank the authors for their detailed response to my previous queries. Several of the responses addressed my previous questions but two key concerns remain.

1. I appreciate that the previous study and data were conceptualized and collected in 2017 before pre-registration became common practice at-large and within the respective labs. Realizing that a post-hoc pre-registration makes little sense, I suggest that the authors report their findings with AND without exclusions to increase transparency. (Posting data upon publication does little to address concerns about robust and replicable findings before publication). For instance, what happens if the 48 participants removed for incomplete CRT scores are included in Study 1a, or the 16 participants excluded from Study 1b for inconsistent social discounting choices are included in analyses?

It is also unclear to me why the authors chose not to pre-register their study design, sample size, measures, and analytic plan for the *new data* collected in Study 2 of the present revision (N=208). As just one example, 111 participants were excluded from this new sample because they failed to pass an attention check. This decision rule could have been documented in advance in a pre-registration to increase confidence in the authors' conclusions. I would encourage the authors to re-run Study 2 with a detailed pre-registration indicating, for instance, the target sample size, exclusion criteria, measurement computations, and analytic strategy.

2. In my last review I suggested that one reason the general population may have difficulty understanding the traits of extraordinary altruists is because they are so rare. To my mind, this is an important alternative explanation that requires data to indicate make the case that this isn't just a broad decision-making error. Or, if it is, what does that mean?

Reviewer #3:

Remarks to the Author:

The authors were very responsive to my and other reviewers' comments and addressed those concerns quite competently. I continue to believe that the present investigation provides unique insights into human prosociality in a way that no previous investigations have done.

I thank the authors for adding Table S7, in which means and SDs of the measures are shown for altruists and controls. This would be particularly useful for other researchers who want to compare the present results against those obtained from other target groups (e.g., population-based or otherwise).

In addition, I believe that the authors have made perceptive points in addressing the concerns of causal direction. It is indeed noteworthy that altruists weren't that much higher in variables representing self-reported altruism (e.g., empathic concern), but instead noticeably higher in self-reported traits of Honesty-Humility, whose scale items do not involve any direct mention of helping or altruistic behavioral tendencies. Again these results would not have been predicted by many experienced personality psychologists, let alone by lay participants, as demonstrated in Study 2 of the present study.

To sum up, I think this is fascinating work offering important new insights into extraordinary altruism.

Thank you very much for considering our manuscript. We appreciate the reviewers' encouraging comments and feedback. We also thank the editor for the opportunity to revise and resubmit our manuscript and for their patience over the last year while we collected the necessary data to address the reviewers' remaining concerns.

We are very pleased to submit our revised manuscript, which now features data from 991 participants in total across five studies our laboratory has conducted (study 1ab; study 2abc), including our initial sample of very rare altruistic participants, a new sample of (non-altruistic) rare populations of people, and a new sample of participants in a pre-registered study that replicates our previous findings. Additionally, we now report analyses of pre-pandemic data from 158,310 U.S. participants (from an international sample of 347,192 participants) who completed the same items from the HEXACO personality inventory that we used (Lee & Ashton 2020; Ashton & Lee, 2018).

Please find our detailed responses to the reviewers below. All new or edited text is marked with **red font** in our main manuscript, supplementary information, and responses to reviewers below.

Reviewer #1 (Remarks to the Author):

Thank you very much for the chance to review the revised version of the manuscript “Unselfish traits and social decision-making patterns characterize six populations of real-world extraordinary altruists”. I think that the authors addressed most of the comments by (the other reviewers and) me in a convincing manner, including collecting new data and providing more information/results.

One comment that I still have is that I find the defense of the comparison sample not overly convincing—although I can see the arguments for not collecting data of a new comparison sample now. For instance, one could have aimed to recruit a larger comparison sample in order to ensure a proper matching; even the “matching on the group level” as argued by the authors has not worked out perfectly. I would thus recommend highlighting this limitation more clearly, especially because there are age and gender (but not educational level) effects for Honesty-Humility, based on the meta-analysis on the HEXACO-PI-R (Moshagen et al., 2019).

Thank you for this suggestion. To address this point and remaining concerns about our control sample, we requested access to a very large dataset of adults (N=347,192) who completed the HEXACO and whose data were reported by Lee & Ashton (2020). This population was suitable because data collection was performed prior to the global pandemic, the study was initially collected using the 100-item measure of the HEXACO and thus all 60 items from the HEXACO-PI-R were available, and information on age and gender were also available. Thus, we ran supplemental analyses using a bootstrapping procedure to draw 5,000 new samples from Lee & Ashton’s population that were age and gender matched to our altruistic group. We found that our control sample did not differ from the distribution of bootstrap means for any of the HEXACO factors. We also replicated our initial finding that Honesty-Humility was the only factor of the HEXACO for which altruists’ mean (and 95% confidence intervals) did not overlap with the distribution of bootstrap means ($p < 0.001$). These findings replicate and, we believe, significantly strengthen our initial conclusions.

In light of these new results, we added the following text to our Results section (pg. 6):

Because our community sample of controls was not perfectly matched to altruists due to a variety of factors (for example, altruists are often selected based on factors related to age, sex, and health and each altruistic group varied widely in their demographic makeup relative to others; see Discussion for further information regarding these limitations), we also sought to confirm whether our finding that Honesty-Humility is the dimension of the HEXACO that most reliably distinguishes real-world extraordinary altruists would replicate in an independent control dataset. To accomplish this, we acquired data from a large population of 347,192 participants who completed the same HEXACO items measured in the present study^{32,33}. We stratified the international dataset by country (United States), age (quantile split), and gender, and randomly drew 5,000 bootstrap samples without replacement that were matched to the full altruistic sample on country, age, and gender. We then compared our altruist sample (N=347) and our initial control sample (N=207) to this new distribution of 5,000 mean scores for each of the HEXACO

personality dimensions. We found that our control sample did not differ from the distribution of bootstrap means for any of the HEXACO dimensions. Even more importantly, we replicated our finding that Honesty-Humility was the only dimension of the HEXACO for which altruists' mean scores (and 95% confidence intervals) did not overlap with the distribution of bootstrap means ($p < 0.001$) (see Supplementary Information, Figure S2).

Supplementary Information, Figure S2:

We also added the following text to our Methods section (pg. 15):

We next tested whether our finding that Honesty-Humility is the dimension of the HEXACO that most reliably distinguishes real-world extraordinary altruists would replicate in an independent control dataset. To accomplish this, we acquired data from a large population of 347,192 participants who completed the same HEXACO items measured in the present study (for recruitment methodology, see Lee & Ashton, 2020). Although participants from Lee & Ashton completed the 100-item version of the HEXACO, we only used their scores computed from the same 60 items in the HEXACO-PI-R. We stratified the international dataset by country (United States), age (quantile split), and gender, and randomly drew 5,000 bootstrap samples without replacement that were matched to the full altruistic sample on each of these demographic variables. Because participants in the Lee & Ashton dataset were heavily skewed younger, we opted for a conservative bootstrap sample size (N=50) to account for potential bias in participant selection during bootstrapping (i.e., we sought to mitigate potential bias for older participants closer to the age of our altruist sample being selected in majority of bootstrap samples). We then compared the altruist sample (N=347) and our initial control sample (N=207) to this new distribution of 5,000 mean scores for each of the HEXACO personality dimensions. Statistical inference was conducted by comparing altruist and control mean scores to the distribution of bootstrap sample means. We calculated the p-value as the proportion of instances in which bootstrap sample means exceeded the mean score for the comparison group.

We also added the following text to our Discussion section and further highlighted our findings within the context of Moshagen et al. (2019) on the effects of age and gender on the HEXACO (pg. 12-13).

Although recent work has found age and gender effects for Honesty-Humility⁷⁵, what group differences emerged (despite our efforts) were statistically controlled for in all analyses. Furthermore, our results are substantiated by supplemental analyses comparing altruists to bootstrapped control samples pooled from a large population sample (N=347,192) that was matched to altruists on country, age, and gender. We found that our initial control sample not differ from the distribution of bootstrap means for any of the HEXACO factors, and also replicated the finding that Honesty-Humility was the only factor of the HEXACO in which altruists' mean (and 95% confidence intervals) did not overlap with the distribution of bootstrap means.

A little bit relatedly, the authors might want to be clearer about the limitation in terms of the temporal order of assessments. Theoretically, one could think about assessing characteristics in a large panel and then evaluating (e.g., after 10, 20 etc. years) who of the panel members has shown acts of extraordinary altruism (obviously, this comes with other limitations; but just to point out that the temporal order was not automatically given).

We agree. We now include this hypothetical assessment in our Discussion to explain what would be required for researchers to be able to draw strong conclusions about the temporal relationship between the traits we measured and altruism (pg. 13).

Temporal conclusions are also limited by the fact that altruistic behaviors were necessarily performed prior to testing. **To draw stronger conclusions that unselfish traits precede altruistic acts in a similarly sized sample of altruists, we would need to collect longitudinal data on our measures in hundreds of thousands of participants and follow them for 10 years or more until a small fraction of them performed real-world extraordinary acts of altruism (e.g., roughly 1 in every 5 million people is recognized as a heroic rescuer, and roughly 17 in every 1 million people is a directed kidney donor; see Supplementary Information Table S1). In lieu of this, we instead aimed to identify measures that distinguish individuals recruited because they had previously engaged in stringently defined real-world altruism. An alternate hypothesis could thus be that altruists' patterns of responding are the result of their altruistic acts. For example, they could have concluded based on their past behavior that they are unusually altruistic or compassionate.**

Another (new) comment that I have is that at the beginning and the end of the abstract, as well as at the beginning of the Introduction, the authors refer to the fact that extraordinary altruism is an evolutionary and motivational puzzle and that, e.g., “self-focused motivations [...] alone are insufficient explanations”. The data, however, do in itself not provide new insights into the evolutionary or motivational reasons for such acts, which readers might expect to learn about. The study is good in describing differences between extraordinary altruists and other people, but the used measures (alone) do hardly help in shedding new light on the underlying (evolutionary or motivational) reasons. Maybe it would be good to adapt corresponding sentences, so that readers do not get wrong expectations and/or to be clear/er what the study adds (and what not). For instance, the first sentence of the Discussion does exactly this (“that generally characterizes them”).

We agree that our data do not speak to the evolutionary basis of costly altruism for strangers, and so have removed references to evolution in the abstract and introduction. We retain references to motivations, because our data include assessments of the subjective value altruists and controls place on outcomes for the self and others, and subjective value is typically defined as the internal value a stimulus has to motivate choices and behavior. We have amended our text to better clarify our goal of identifying the characteristics that distinguish real-world extraordinary altruists from typical people.

For example (pg. 3):

Although insightful research on this topic has emerged in recent years, the underpinnings of real-world acts of extraordinary altruism—for example, non-directed organ donations, heroic rescues, and risky humanitarian aid work—remain a puzzle¹⁻³.

And in the Discussion (pg. 11):

These findings position social discounting patterns as meaningful, replicable, non-self-report-based indices of individual variation in ecologically valid, extraordinary forms of altruism. In that social discounting decisions directly reflect variation in the subjective value of outcomes for various others versus the self, they are valuable for understanding the basis of extraordinary altruism—making large sacrifices or taking significant risks for more distant others is inherently more extraordinary because it is much rarer across all populations. **In that subjective value is typically defined as the internal value a stimulus has to motivate choices and behavior⁶⁰, these findings may help to understand motivations underlying extraordinary altruism.**

A smaller comment that I have is that previous research has aimed to link five-factor model (or Big Five) Agreeableness to altruism, whereas the present study linked HEXACO Agreeableness vs Anger to altruism. There are some noteworthy differences between FFM/Big Five and HEXACO Agreeableness (also depending on the FFM/Big Five measure used). Thus, proponents of the five-factor (vs. HEXACO) approach could argue that this study is not automatically in contrast to previous findings (looking at “not-extraordinary” altruism), but this is implied sometimes (e.g., in the Discussion “previously linked to altruism, including self-reported empathy, agreeableness...”). The authors might want to be more precise about this. On a minor note, my previous comment concerning the interstitial altruism scale in the HEXACO-PI-R was not meant to imply that this measure is “flawed”, but rather that I was surprised that this measure (or any other measure aiming to assess a construct termed “altruism”) was not used. For instance, one could have just added the four (our eight) altruism-items from the HEXACO-PI-R to the HEXACO-60.

We agree on the differences between FFM/Big Five and HEXACO Agreeableness, which have been the subject of extensive discussion. We also agree that clarity and precision on these points is important. Where relevant, we now note throughout the manuscript that HEXACO agreeableness is not identical to five-factor inventory agreeableness so that it is clear that our conclusions only extend to Agreeableness as measured by the HEXACO.

For example, in our Results section (pg. 8):

Notably absent were predictors linked to altruism in prior research, including self-reported empathy⁴⁻⁶, agreeableness⁷ (**although note that the agreeableness dimension of the HEXACO is not identical to this dimension in five-factor inventories**), or temperamental risk-perceptions/boldness⁸.

Smaller points:

- Maybe use “expect” than “recognize” on line 361?

We have amended this.

- I don’t understand what is meant by “the right information at the right time – a response consistent with the high Honesty-Humility scores” (line 451/452). HH scores do not say anything about which information people have (I think)?

We apologize for any confusion. The phrase “*they consistently report that they are not especially altruistic or unusual in any way, but simply had the right information at the right time*” was in reference to altruists describing their acts of altruism in ways that are consistent with trait Honesty-Humility, but is not drawn from the Honesty-Humility measure itself. To make this clearer, we have amended this sentence to read (pg. 13):

First, altruists do not score highly on self-report altruism scales that are more straightforwardly related to altruism^{21,76} (including empathic concern in the present study) and in interviews they consistently report that they are not especially altruistic or unusual in any way, but **simply acted as anyone with the same information and opportunity would have^{18,77}. This interview response is most** consistent with the high Honesty-Humility scores observed in the present study.

Reviewer #2 (Remarks to the Author):

I appreciate the chance to review this manuscript again and I thank the authors for their detailed response to my previous queries. Several of the responses addressed my previous questions but two key concerns remain.

1. I appreciate that the previous study and data were conceptualized and collected in 2017 before pre-registration became common practice at-large and within the respective labs. Realizing that a post-hoc pre-registration makes little sense, I suggest that the authors report their findings with AND without exclusions to increase transparency. (Posting data upon publication does little to address concerns about robust and replicable findings before publication). For instance, what happens if the 48 participants removed for incomplete CRT scores are included in Study 1a, or the 16 participants excluded from Study 1b for inconsistent social discounting choices are included in analyses?

We would like to emphasize that all of our analyses included all usable data. For example, all findings reported in the previous submission of our manuscript include the referenced 48 participants and 16 participants in Study 1a where possible. We did not exclude these participants from the entire study as the reviewer suggests. We only excluded 48 participants from analyses that included the CRT because they did not perform this task, but these 48 participants who had complete data for other measures were indeed included in other analyses (both Study 1a and 1b). We only excluded 16 participants who had switched back and forth multiple times during the social discounting task because an indifference point (the key dependent variable in this task) is not computable from their data. But these 16 participants who had complete data for other measures were indeed included in our analyses for Study 1a.

We previously outlined this in our Methods section: “One control participant did not complete the HEXACO and was list-wise excluded for analyses including this measure. 48 participants did not complete the cognitive reflection test (CRT) and were list-wise excluded for analyses including this measure (45 non-directed kidney donors, 3 heroic rescuers).”

We now include the additional text in red to make this clear (pg. 14):

One control participant did not complete the HEXACO and was list-wise excluded for analyses including this measure. 48 participants did not **perform** the cognitive reflection test (CRT) and were list-wise excluded for analyses including this measure (45 non-directed kidney donors, 3 heroic rescuers). **The participants excluded from HEXACO and CRT analyses were included in all other analyses if they had complete data for the measures used in those analyses. In other words, all usable data was included in all analyses.**

And on pg. 15 for social discounting:

The participants excluded from social discounting analyses were included in all other analyses if they had complete data for the measures used in those analyses. In other words, all usable data were included in all analyses.

It is also unclear to me why the authors chose not to pre-register their study design, sample size, measures, and analytic plan for the *new data* collected in Study 2 of the present revision (N=208). As just one example, 111 participants were excluded from this new sample because they failed to pass an attention check. This decision rule could have been documented in advance in a pre-registration to increase confidence in the authors' conclusions. I would encourage the authors to re-run Study 2 with a detailed pre-registration indicating, for instance, the target sample size, exclusion criteria, measurement computations, and analytic strategy.

At the reviewer's request, we re-ran this study for the third time and outlined our target sample size, exclusion criteria, measurement computations, and analytic strategy in a pre-registration. Our pre-registration followed the AsPredicted template and is available on the Open Science Framework: https://osf.io/7t4qf/?view_only=a1072f3df69b46dc9af203a63c4858fa

We now report this as a new confirmatory study (Study 2b) in our manuscript and describe our findings that are consistent with the data from the previous sample (Study 2a) in our Results section (pg. 8-9).

We also aimed to assess whether beliefs about of altruists in the general population were consistent with the personality traits and decision-making patterns we observed among actual altruists. **We conducted this assessment in an exploratory study (N=208) and a pre-registered confirmatory study (N=201; pre-registration link: https://osf.io/7t4qf/?view_only=a1072f3df69b46dc9af203a63c4858fa).** Each study recruited a sample of American adults using a Qualtrics panel designed to match census-based population demographics in terms of gender, age, race and ethnicity, and education (see Supplementary Information, Table S9).

Procedures were similar across both studies. Participants were asked to consider how an individual representing each of the six groups of altruists we assessed compared to the average person in terms of altruism (i.e., how altruistic they perceived each to be), risk (i.e., how risky they perceived each type of altruism to be), and the six HEXACO dimensions (detailed explanations of all traits were provided **for each study**; see Supplementary Information, Table S10a-b).

All participants also completed a third-person social discounting task, which assessed their judgments about how an individual who had engaged in each of the six forms of altruism (as well as an average person) would be likely to allocate resources for others (see Methods). These ratings directly represented beliefs about the "amount each target would be willing to forgo" ($v_{beliefs}$), such that estimated social discounting rates ($logk_{beliefs}$) could be calculated using the hyperbolic discounting model separately for each group for each participant.

Across both the exploratory and confirmatory studies, linear mixed-effects modeling revealed that respondents perceive all forms of extraordinary altruism to be both relatively altruistic and risky (see Supplementary Information, Table S11a-b). **Both samples** also judged altruists as differing from the average person in terms of all six

HEXACO personality dimensions. Estimated social discounting rates corresponding to judgments about how much an individual is willing to forgo for various social others (*logkbeliefs*) were shallower for all altruistic groups relative to those estimated about the average person. Thus, respondents correctly viewed altruists as more willing to forgo resources for others at varying social distances (social discounting)—linking altruism to unselfish behavior. However, they did not specifically associate extraordinary altruism with traits related to unselfishness. Rather, they associated extraordinary altruism with undifferentiated traits broadly construed as positive, **thus predicting 1 out of 6 personality dimensions correctly** (see Supplementary Information, Figure S4a-e).

And in our Methods section (pg. 20-21):

Study 2b (Pre-registered)

To confirm our findings, we pre-registered (pre-registration link: https://osf.io/7t4qf/?view_only=a1072f3df69b46dc9af203a63c4858fa) and conducted a second study 2b. Procedures were carried out in accordance with a protocol approved by the Institutional Review Board at Linfield College in McMinnville, Oregon, and all participants provided electronic written informed consent upon enrollment.

Participants. We again aimed to recruit a sample of 200 American adults using a Qualtrics panel designed to match census-based U.S. population demographics: gender (51% Female), age (30.5% 18-34 years, 34.4% 35-54 years, 35.2% 55+ years), race/ethnicity (62.3% non-Hispanic white, 12.4% non-Hispanic black, 17.3% Hispanic, 5.4% Asian, 2.6% other), and education (24% HS degree or less, 48% Bachelors or some college, 28% graduate or professional degree). We recruited this representative sample while concurrently accounting for participants that failed to pass the inclusion criteria we explicitly outlined in the pre-registration. A total of 201 participants were recruited by Qualtrics from a representative panel of potential U.S. participants approximately corresponded to target demographics in terms of gender (50.25% female, 50.25% male, 0.5% other), age (30.85% 18-34 years, 33.83% 35-54 years, 35.32% 55+ years), race/ethnicity (60.2% non-Hispanic white, 12.44% non-Hispanic black, 16.92% Hispanic or Latino, 4.98% Asian, 5.47% other), and education (23.88% HS degree or less, 47.76% Bachelors or some college, 28.36% graduate or professional degree).

Procedure. Our procedures followed those outlined in Study 2a and our pre-registration. To assess perceptions of altruists and their respective prosocial behaviors, we asked an independent sample of participants to rate various individuals in terms of nine dimensions: altruism, risk, the six HEXACO dimensions, and social discounting. Participants provided their perceptions of the average person, of exemplars of each of the types of six types of altruists that were the focus of our study (liver donors, heroic rescuers, non-directed kidney donors, humanitarian aid workers, directed kidney donors, and bone marrow donors), and six groups who are notable for non-altruistic actions included to minimize participants' ability to discern the focus of the study (internet trolls, tax evaders, marathon runners, analog astronauts, former contestants in the national Miss America pageant, and BASE jumpers). For each dimension, groups were displayed in

random order and rated using 5-point scales. Definitions of each trait and descriptions of each group were provided (see Supplementary Information, Table S10a-b).

Participants also completed a third-person social discounting task, which assessed judgments about how an individual who had engaged in each of the six forms of altruism (as well as an average person) would be likely to allocate resources for others. Participants were instructed to imagine that [an individual representing each group] made a list of 100 people closest to him or her in the world ranging from their closest friend or relative at #1 to a mere acquaintance or stranger at #100. They were then instructed to move a slider bar "to indicate how much of their resources this person would sacrifice for each of the following people." They were presented with seven slider bars (for person #1, #2, #5, #10, #20, #50, and #100) which could be moved from 0 (sacrifice no resources; keep all for self) to 100 (sacrifice all resources; give all away). These ratings directly represented judgments about the amount the target would be willing to forgo ($v_{beliefs}$). Social discounting rates ($logk_{beliefs}$) were estimated using the hyperbolic discounting model separately for each group for each participant.

Statistical Analyses. To assess how participants rated each type of altruist in comparison to the average person on each characteristic, we employed a series of linear mixed-effects regressions using the `lme` function within the `nlme` package in R version 3.6.3, which allowed intercepts to vary across participants. In each model, we entered groups as indicator variables with "average person" as the baseline group.

2. In my last review I suggested that one reason the general population may have difficulty understanding the traits of extraordinary altruists is because they are so rare. To my mind, this is an important alternative explanation that requires data to indicate make the case that this isn't just a broad decision-making error. Or, if it is, what does that mean?

At the reviewer's request, we spent the last year recruiting samples of two additional rare groups we selected because they have a known prevalence rate that is less than or equal to our rare altruistic groups and because they are not characterized by unusual altruistic behavior to shed some light on this interesting alternative hypothesis. We added the following text in our Results section (pg. 9-10):

To explore whether this prediction error was specific to altruism, or whether it was simply a function of extraordinary altruists' rarity, we also recruited two independent samples drawn from equally rare populations (N=28) that are not defined by altruism. They included a sample of extreme athletes (BASE jumpers)⁴⁶⁻⁴⁹ and former contestants in the national Miss America pageant. All completed the battery of HEXACO-PI-R items. These groups were selected because their prevalence rate can be estimated and are similar to that of our altruistic groups. There have been roughly 2,500 BASE Jumpers worldwide since 1981 as estimated by basenumbers.org as of August 2022 (prevalence assuming all are alive today: 0.0000075%)⁵⁰, there are 51 Miss America contestants per year (prevalence assuming all are alive today since the first pageant in 1921: 0.000015%). Furthermore, the annual fatality rate of BASE jumping is higher than that of organ donation, estimated to be 1 death for every 60 jumpers and the serious injury rate

(requiring hospital care) as 0.2–0.4 % per jump (i.e., a 5- to 16-fold risk for death or injury when compared with skydiving)⁴⁹.

Using the bootstrapping procedure described above, we then compared each of these new rare groups to a distribution of 5,000 bootstrap samples that were matched to each group on country (United States), age (quantile split), and gender (female only for Miss America contestants). We found that Miss America contestants on average scored lower in honesty-humility, higher in extraversion, and higher in conscientiousness compared to the matched distribution of bootstrap means (see Supplementary Information, Figure S5a). BASE jumpers on average scored lower in emotionality, higher in agreeableness, and lower in openness to experience compared to the matched distribution of bootstrap means (see Supplementary Information, Figure S5b).

We then tested whether the general population from the second sample of participants (N=201) would uniquely link each group to these characteristics. We found that the general population was broadly accurate at predicting Miss America contestants' traits (correctly predicting 5 out of 6 personality dimensions), but were less accurate at predicting BASE Jumpers' traits (correctly predicting 2 out of 6 personality dimensions) (see Supplementary Information, Table S12). This finding provides preliminary evidence that the general population's prediction errors for altruists (who they misjudged on 5 out of 6 traits) might be specifically related to misconceptions of altruism and not simply reflect altruistic groups being so rare. This finding also suggests that the general population's responses regarding altruists did not result from a broad decision-making error because participants' beliefs were broadly accurate for Miss America contestants.

Discussion section (pg. 10):

But, despite compelling empirical evidence specifically linking real-world altruism and unselfishness, **two independent representative samples** of American adults did not specifically associate extraordinary altruism with traits related to unselfishness, but rather with undifferentiated traits broadly construed as positive but not closely related to actual altruism, including high extraversion, high agreeableness, and high conscientiousness. (Furthermore, this prediction error seemed to be relatively specific to altruism, as the general population was more accurate at predicting traits characterizing similarly rare groups who were not defined by altruistic actions.)

Discussion section (pg. 12):

It could be argued that the rarity of these forms of altruism may have made it more difficult for our population sample to predict what traits would distinguish altruists, **although their greater accuracy in predicting the personality traits of BASE jumpers and Miss America contestants (who are similarly rare) contradicts this possibility.**

Methods section (pg. 21-22):

Study 2c

To explore whether this prediction error on behalf of the general population was specific to altruism, or whether it could be due to their rarity, we conducted a study in which we recruited two independent samples from rare populations. All study 2c procedures were carried out in accordance with a protocol approved by the Institutional Review Board at Georgetown University, Washington DC, and all participants provided electronic written informed consent upon enrollment.

Participants. We recruited two independent samples of rare populations (N=28), including extreme athletes (e.g., BASE Jumpers) and former contestants in the national Miss America pageant. Recruitment of BASE Jumpers was carried out using previous recruitment pools from researchers who have assessed personality traits of these athletes⁴⁶⁻⁴⁹ as well as online public forums (e.g., <https://www.blincmagazine.com/forum/>). Recruitment of Miss America contestants was carried out via snowball sampling and via online forums (e.g., Facebook groups).

Bootstrap sampling. To generate demographically-matched control samples for each of these new rare groups, we used the bootstrapping procedure described above. We again used data from a large population of 347,192 participants who completed the same HEXACO items measured in the present study (for recruitment methodology, see Lee & Ashton, 2020). Although participants from Lee & Ashton completed the 100-item version of the HEXACO, we only used their scores computed from the same 60 items in the HEXACO-PI-R. We stratified the international dataset by country (United States), age (quantile split), and gender (female only for Miss America contestants), and randomly drew 5,000 bootstrap samples without replacement that were matched on each of these demographic variables to the Miss America contestant and BASE Jumper samples, respectively. Because participants in the Lee & Ashton dataset were heavily skewed younger, we again opted for the same conservative bootstrap sample size (N=50) to account for potential bias in participant selection during bootstrapping (i.e., we sought to mitigate potential bias for older participants closer to the age of our rare samples being selected in majority of bootstrap samples). We then compared the Miss America contestant and BASE Jumper samples to their respective bootstrap distribution of 5,000 control mean scores for each of the HEXACO personality dimensions. Statistical inference was conducted by comparing Miss America contestant and BASE Jumper mean scores to their respective distribution of bootstrap sample control means. We calculated the *p*-value as the proportion of instances in which bootstrap sample means exceeded the mean score for the comparison group.

Statistical Analyses. We then tested whether the general population from the second sample of participants (N=201; Study 2b) would uniquely link these each group to these characteristics. To assess how participants in the general population rated each rare group in comparison to the average person on each characteristic, we employed a series of linear mixed-effects regressions using the `lme` function within the `nlme` package in R version 3.6.3, which allowed intercepts to vary across participants. In each model, we entered groups as indicator variables with "average person" as the baseline group.

Reviewer #3 (Remarks to the Author):

The authors were very responsive to my and other reviewers' comments and addressed those concerns quite competently. I continue to believe that the present investigation provides unique insights into human prosociality in a way that no previous investigations have done.

I thank the authors for adding Table S7, in which means and SDs of the measures are shown for altruists and controls. This would be particularly useful for other researchers who want to compare the present results against those obtained from other target groups (e.g., population-based or otherwise).

In addition, I believe that the authors have made perceptive points in addressing the concerns of causal direction. It is indeed noteworthy that altruists weren't that much higher in variables representing self-reported altruism (e.g., empathic concern), but instead noticeably higher in self-reported traits of Honesty-Humility, whose scale items do not involve any direct mention of helping or altruistic behavioral tendencies. Again these results would not have been predicted by many experienced personality psychologists, let alone by lay participants, as demonstrated in Study 2 of the present study.

To sum up, I think this is fascinating work offering important new insights into extraordinary altruism.

We are overjoyed that this reviewer believes our investigation provides unique insights into human prosociality in a way that no previous investigations have. We sincerely thank the reviewer for their time and feedback.

Reviewer 1 additional comments on response to Reviewer 4:

I will first say something about each point raised by Reviewer #4, and then also make a final comment by me (in addition to my review):

The primary challenge is that the data are conditioned people having *already* engaged in acts of extraordinary altruism. I have difficulty believing that this constellation of traits/preferences is discriminantly predictive of extraordinary altruism. Sure, once I've given a kidney or rescued someone from a burning building, I see myself as someone who values others over myself (and report as such on personality scales). The problem is that there are likely MANY more people who score high on H, low on PD, and have shallow social discounting rates than there are people giving up kidneys or running into burning buildings (though I'd be unsurprised to learn that they're all nurses, public school teachers, and social workers). So what is actually doing the work in the moment that people make these kinds of extraordinary decisions? Or as the authors ask in the first paragraph "what generally distinguishes individuals who engage in stringently defined real-world altruism is unselfishness?" We can't answer this question with these data.

I think that the authors made it clearer what their research is about and what we can (not) learn from it, although I also think that they do not describe the limitations of their study that clearly that all readers understand them directly. The question (by the reviewer) of whether the found "constellation of traits/preferences is discriminantly predictive of extraordinary altruism" cannot be answered with the data and the authors do not clearly mention this and do also not clearly provide a suggestion how this could be tested (e.g., via a panel followed over many years and then looking who of those showed acts of extraordinary altruism over the time).

One potential suggestion to further tackle this concern by the review is to use past tense when describing what can be learnt from the study, like traits that distinguish people who showed acts of extraordinary altruism from those who did—when being tested—not.

Thank you for Reviewer 1's comments for this section.

We now state clearly throughout the manuscript that our findings can help us understand traits and preferences that distinguish people who have already engaged in acts of extraordinary altruism and are not aimed at prospectively predicting extraordinary altruism. In response to Reviewer 3, we also describe what would be required to reverse the temporal ordering of our measures (pg. 13):

Temporal conclusions are also limited by the fact that altruistic behaviors were necessarily performed prior to testing. **To draw stronger conclusions that unselfish traits precede altruistic acts in a similarly sized sample of altruists, we would need to collect longitudinal data on our measures in hundreds of thousands or millions of participants and follow them for 10 years or more until a small fraction of them performed real-world**

extraordinary acts of altruism (e.g., roughly 1 in every 5 million people is recognized as a heroic rescuer, and roughly 17 in every 1 million people is a directed kidney donor; see Supplementary Information Table S1). In lieu of this, we instead aimed to identify measures that distinguish individuals recruited because they had previously engaged in stringently defined real-world altruism. An alternate hypothesis could thus be that altruists' patterns of responding are the result of their altruistic acts. For example, they could have concluded based on their past behavior that they are unusually altruistic or compassionate.

The reviewer is also correct that not all people with certain levels in the identified traits showed, show, and will show extraordinary altruism, which could also be said more clearly by the authors. However, this is quite trivial in terms of that this is how corresponding research works – and people with such trait levels will, if one interprets the presented data in this way, be more likely to show acts of extraordinary altruism.

“A widespread assumption across the social and biological sciences and among the general public is that selfish motivations underlie all human behavior, including prosocial behaviors¹⁻⁴.” With all due respect, this is a strawman. Social and biological sciences are not universally, nor are all their theories Hobbesian. In fact several of the authors on this paper have made this point very persuasively over the last decade if not longer! This framing mischaracterizes the literature and undermines the paper.

p. 5 typo: “We conducted a conducted a series of...”

I think this is dealt with properly.

“Because our groups differed in average gender distribution and age “ Why not compare them to age- and gender-matched controls? Surely it would be easy to administer these batteries (e.g., online) to a large control panel and then match from there.

This was also a concern by other reviewers, and although I can see the arguments why the authors did not collect new data, it is still not a very good control group that they recruited (in 2017). One could say: The authors defend the data that they collected as good as they can (and also provide several analyses supporting their defense), but I would also still say that the control group is just not very good.

As we describe above in response to your review, to address lingering concerns about our control sample, we requested access to a very large dataset of U.S. adults (N=347,192) who completed the HEXACO, and whose data were reported by Lee & Ashton (2020). This population was suitable because data collection was performed prior to the global pandemic, the study was initially collected using the 100-item measure of the HEXACO and thus all 60 items from the HEXACO-PI-R were available, and information on age and gender were also available. Thus, we ran supplemental analyses using a bootstrapping procedure to

draw 5,000 new samples from Lee & Ashton's population that were age and gender matched to our altruistic group. We found that our control sample did not differ from the distribution of bootstrap means for any of the HEXACO factors. We also replicated our initial finding that Honesty-Humility was the only factor of the HEXACO for which altruists' mean (and 95% confidence intervals) did not overlap with the distribution of bootstrap means ($p < 0.001$). These findings replicate and, we believe, significantly strengthen our initial conclusions.

“For every unit increase in Honesty-Humility score, participants were 83.47% more likely to be altruists” Is this a typo? This is one of those findings that would make me go back and double check the analyses. If this were true we'd expect to have much shorter kidney waitlists than we currently have.

I think that their answer is good, but they might want to add the argumentation from the rebuttal letter which is not in the manuscript: “It's important to note that because the base rate of these altruistic behaviors is so low (as we detail in our manuscript) even an 83.74% increase in that base rate still yields a low value”. (of course, adapted with regard to that they now describe how much more likely it is to be an altruist).

We now include the following in our manuscript (pg. 8):

Note also that because the base rate of these altruistic behaviors is so low even an 84% increase in that base rate still yields a low value.

“However, they did not perceive that altruists are distinguished by unselfish traits (Honesty-Humility) or behavior (social discounting)” For what it is worth I wouldn't have thought honesty-humility is associated with being unselfish either because those words mean different things. (In fact reading this paper inspired me to go look at the H items and my conclusion is that Honesty/Humility is a poor label for that factor; it's really more like egalitarianism or universalism.) As for the social distancing null, this concerns me greatly because if anything this item should have acted a *comprehension check* in these data. Participants didn't think a person who sacrificed their safety/well-being/comfort to help a stranger would also be more likely to give up money for a stranger than a control??

I think this is dealt with properly.

“These findings position social discounting patterns as a meaningful, replicable, non-selfreport-based index of individual variation in ecologically valid altruism.” I don't see how one can draw this conclusion. For example, conservatives are MORE compassionate and giving to those close to them relative to liberals, whereas liberals have flatter curves and are more generous to those farther from them (Waytz et al., 2019; Enke et al., 2019) but how can you determine which is more altruistic? Wouldn't it make more sense to fix the target and then administer the Social Value Orientation measure to determine who is more altruistic at each level?

I think this is dealt with properly.

Reviewer 3 additional comments on response to Reviewer 4:

I have now reviewed the reviewer' points and the authors' replies to them. Here are my thought on those points.

I believe Reviewer #4 had two main issues. First, he/she questioned whether the current study can answer questions such as “what is actually doing the work in the moment that people make these kinds of extraordinary decisions”. I think it was quite clear that the current research was not meant to answer questions of this nature. The reviewer also stated that “The problem is that there are likely MANY more people who score high on H, low on PD, and have shallow social discounting rates than there are people giving up kidneys or running into burning buildings”. This statement is undoubtedly true, but there is no contraction between this fact and the current finding. I believe that the authors clarified this point adequately in the letter, and I think it's a good decision not to address this point within the article itself.

The second point of Reviewer #4 is about causal direction, which is an important issue to be addressed. There is no practical way to address this issue empirically because the current study examines extremely low-base rate behaviors; that is, a prospective longitudinal study can hardly be a practical research design option. The authors acknowledged this limitation adequately in the article, but they also provided a highly persuasive point as to why the present findings can hardly be explained by the effects of previous prosocial experiences on people's self-perception. The altruists in the present research did not show noticeably higher scores on the self-reported scales assessing general altruism (e.g., empathic concern). Instead they showed significantly higher (or lower) scores on measures/scales that do not involve any direct mention about “altruism/helping” such as Honesty-Humility, Personal Distress, and Social Distance Rates.

The remaining points raised by Reviewer #4 were helpful and relatively minor ones, and the authors revised their paper following those suggestions. Most of these revisions are uncontroversial and adequately addressed in my opinion.

I may have one point to make about Study 2—this had not caught my attention before (sorry), but it did while reviewing the results from Study 2, especially those reported in a new supplementary table (Table S11). The results indicated that participants were unable to correctly identify the individual differences variables that actually distinguish altruists from non-altruists. Rather they perceived the extreme altruists to be high on all of the attributes that are generally perceived to be positive. I think the fact that the participants provided rather undifferentiated ratings (i.e., overall positive ratings) may indicate that they did not commit to the rating task as hard as they could. This is somewhat plausible considering that the rating task might have been cognitively arduous. Although I don't think this undermines the significance of the major findings of the present research, it might be prudent to mention this as one of the Study 2 limitations.

Thank you for Reviewer 3's comments for this section.

Our new confirmatory data from our pre-registered Study 2b address the new concern regarding overall positive ratings. This independent sample also rates altruists' traits as being overall positive but does not produce similar positive ratings for our other rare groups. For example, they rate Miss America contestants lower in Honesty-Humility on average and rate BASE Jumpers lower in agreeableness on average.

Reviewers' Comments:

Reviewer #1:

Remarks to the Author:

Thank you very much for the chance to review the further revised version of the manuscript "Unselfish traits and social decision-making patterns characterize six populations of real-world extraordinary altruists". Again, I think that the authors addressed most of the comments by (the other reviewers and) me in a convincing manner, including collecting new data and providing more information/results. I just have very few very minor comments left (which might also rather be a question of personal style):

- "whether real-world extraordinary altruists are actually distinguished by these factors has not been tested".

-> maybe add "empirically" before tested

- "are most predictive of behavioral outcomes in novel contexts or those lacking strong norms"

-> I would suggest rephrasing to "in novel contexts lacking strong norms" (because in novel contexts with strong norms, stable trait-level differences might not be most predictive)

- "The rarity of extraordinary acts like heroic rescues and altruistic organ donations renders contexts in which these acts occur novel and lacking strong norms by default, and thus particularly likely to correspond to dispositional variation."

-> I think here could be a difference between descriptive and injunctive social norms (I am not sure either, but it appears to me as if the case is a little bit clearer for descriptive vs. injunctive social norms). More generally, though, I think the better theoretical framework could be derived from trait-activation theory (e.g., De Vries et al., 2016, EHB), suggesting that such situations allow people with high levels in unselfishness/prosociality to express their tendencies, instead of simply referring to new situations without norms.

- Given that you now (and thanks for this!) refer to differences between HEXACO vs. FFM/Big Five Agreeableness, you might want to direct readers to a recent meta-analysis on this:

<https://doi.org/10.1177/08902070211026793>

- Most limitations are mentioned briefly, accompanied by a typically longer explanation of why this is not so problematic. This is probably a matter of style, but especially with regard to the sample sizes, I would prefer a clearer statement that the sample sizes were in absolute terms quite low, which might have impacted detecting a true effect (prior to the explanation why this might not be so problematic at all).

That all being written, I really enjoyed reading the manuscript.

Reviewer #2:

Remarks to the Author:

It is my third time reading this paper. I am very grateful to the authors for responding to my previous feedback by taking the time to add pre-registered tests of their key question and new samples (e.g., BASE jumpers and former Ms. American contestants) to address a likely possible alternative explanation. I am fully satisfied with this thorough and thoughtful revision, and I think the authors for their important contribution to the literature.

Reviewer #3:

Remarks to the Author:

The authors have been very responsive to the comments raised by the reviewers, and as a result, the manuscript has improved considerably again. Collecting new data and using a new large control group were very useful in addressing some of the reviewers' concerns.

I have only one comment to provide for the revised manuscript. In Figure S2, it says "Means and SEs are listed in Table S7". But Table S7 does not include bootstrap means (and SDs?) for the HEXACO variables obtained from the new control group. (In addition, shouldn't "SEs" be "SDs"?)

Thank you for the opportunity revise our manuscript one last time to address the remaining concerns of the reviewers and editorial requests. Please see our responses and relevant changes below.

Reviewer #1 (Remarks to the Author):

Thank you very much for the chance to review the further revised version of the manuscript “Unselfish traits and social decision-making patterns characterize six populations of real-world extraordinary altruists”. Again, I think that the authors addressed most of the comments by (the other reviewers and) me in a convincing manner, including collecting new data and providing more information/results. I just have very few very minor comments left (which might also rather be a question of personal style):

- “whether real-world extraordinary altruists are actually distinguished by these factors has not been tested”.

-> maybe add “empirically” before tested

We added “empirically” before tested (pg. 3).

- “are most predictive of behavioral outcomes in novel contexts or those lacking strong norms”

-> I would suggest rephrasing to “in novel contexts lacking strong norms” (because in novel contexts with strong norms, stable trait-level differences might not be most predictive)

We rephrased to read “in novel contexts lacking strong norms” (pg. 3)

- “The rarity of extraordinary acts like heroic rescues and altruistic organ donations renders contexts in which these acts occur novel and lacking strong norms by default, and thus particularly likely to correspond to dispositional variation.”

-> I think here could be a difference between descriptive and injunctive social norms (I am not sure either, but it appears to me as if the case is a little bit clearer for descriptive vs. injunctive social norms). More generally, though, I think the better theoretical framework could be derived from trait-activation theory (e.g., De Vries et al., 2016, EHB), suggesting that such situations allow people with high levels in unselfishness/prosociality to express their tendencies, instead of simply referring to new situations without norms.

We agree this is an interesting consideration. However, it appears that De Vries et al. (2016) primarily focuses on kin-based and reciprocal altruism (e.g., Table 2), which is somewhat outside the purview of our study.

De Vries, R. E., Tybur, J. M., Pollet, T. V., & Van Vugt, M. (2016). Evolution, situational affordances, and the HEXACO model of personality. *Evolution and human behavior*, 37(5), 407-421.

- Given that you now (and thanks for this!) refer to differences between HEXACO vs. FFM/Big Five Agreeableness, you might want to direct readers to a recent meta-analysis on this: <https://doi.org/10.1177/08902070211026793>

We now cite this meta-analysis (pg. 5).

- Most limitations are mentioned briefly, accompanied by a typically longer explanation of why this is not so problematic. This is probably a matter of style, but especially with regard to the sample sizes, I would prefer a clearer statement that the sample sizes were in absolute terms quite low, which might have impacted detecting a true effect (prior to the explanation why this might not be so problematic at all).

We now add the following sentence to our limitations section on page 12: “Thus, the sample sizes were low, which might have affected the detection of true effects.”

That all being written, I really enjoyed reading the manuscript.

Reviewer #2 (Remarks to the Author):

It is my third time reading this paper. I am very grateful to the authors for responding to my previous feedback by taking the time to add pre-registered tests of their key question and new samples (e.g., BASE jumpers and former Ms. American contestants) to address a likely possible alternative explanation. I am fully satisfied with this thorough and thoughtful revision, and I think the authors for their important contribution to the literature.

Reviewer #3 (Remarks to the Author):

The authors have been very responsive to the comments raised by the reviewers, and as a result, the manuscript has improved considerably again. Collecting new data and using a new large control group were very useful in addressing some of the reviewers’ concerns.

I have only one comment to provide for the revised manuscript. In Figure S2, it says “Means and SEs are listed in Table S7”. But Table S7 does not include bootstrap means (and SDs?) for the HEXACO variables obtained from the new control group. (In addition, shouldn’t “SEs” be “SDs”?)

Thank you for pointing this out. We now clarify that “Means and SDs for altruists and controls are listed in Table S7”